# GLOBGM v1.0: a parallel implementation of a 30 arcsec PCR-GLOBWB-MODFLOW global-scale groundwater model

Jarno Verkaik[1,2], Edwin H. Sutanudjaja[1], Gualbert H.P. Oude Essink[1,2], Hai Xiang Lin[3,4], Marc F.P. Bierkens[2,1]

[1]Unit Subsurface and Groundwater Systems, Deltares, Utrecht, the Netherlands
[2]Department of Physical Geography, Faculty of Geosciences, Utrecht University, Utrecht, the Netherlands
[3]Department of Applied Mathematical Analysis, Faculty of Electrical Engineering, Mathematics and Computer Science, Delft University of Technology, Delft, the Netherlands
[4]Institute of Environmental Sciences, Faculty of Science, Leiden University, Leiden, the Netherlands

*Correspondence to*: Jarno Verkaik (Jarno.Verkaik@deltares.nl)

## Abstract

We discuss the various performance aspects of parallelizing our transient global-scale groundwater model at 30″ resolution (30 arcseconds; ~1 km at the equator) on large, distributed memory parallel clusters. This model, referred to as the GLOBGM, is the successor of our 5′ (5 arcminutes; ~10 km at the equator) PCR-GLOBWB 2 groundwater model based on MODFLOW

having two model layers. The current version of the GLOBGM (v1.0) used in this study also has two model layers, is uncalibrated, and uses available 30″ PCR-GLOBWB data. Increasing the model resolution from 5′ to 30″ creates challenges including increased runtime, memory usage and data storage, that exceed the capacity of a single computer. We show that our parallelization tackles these problems with relatively low parallel hardware requirements to meet users/modelers who do not have exclusive access to hundreds or thousands of nodes within a supercomputer.

For our simulation we use unstructured grids and a prototype version of MODFLOW 6 that we have parallelized using the message-passing interface. We construct independent unstructured grids having a total of 278 million active cells to cancel all redundant sea and land cells, while satisfying all necessary boundary conditions, and distribute them over three continental-scale groundwater models (Afro-Eurasia; 168 M, Americas; 77 M, and Australia; 16 M) and one remainder model for the smaller islands (17 M). Each of the four groundwater models is partitioned into multiple non-overlapping submodels that are

tightly coupled within the MODFLOW linear solver, where each submodel is uniquely assigned to one processor core and associated submodel data is written in parallel during the pre-processing using data tiles. For balancing the parallel workload in advance, we apply the widely used METIS graph partitioner in two ways: straightforwardly applied to all (lateral) model grid cells, and area-based applied to HydroBASINS catchments that are assigned to submodels for pre-sorting to a future coupling with surface water. We consider an experiment for simulating 1958-2015 with daily timesteps and monthly input,

including a 20-year spin-up, on the Dutch national supercomputer Snellius. Given that the serial simulation would require ~4.5 months of runtime, we set a hypothetical target of a maximum of 16 hours of simulation runtime. We show that 12 nodes (32 cores per node, 384 cores in total) are sufficient to achieve this target, resulting in a speed-up of 138 for the largest Afro-Eurasia model using 7 nodes (224 cores) in parallel.

A limited evaluation of the model output using NWIS head observations for the contiguous United States was conducted.

This showed that increasing the resolution from 5′ to 30″ results in a significant improvement with GLOBGM for the steady-state simulation compared to the 5′ PCR-GLOBWB groundwater model. However, results for the transient simulation are quite similar and there is much room for improvement. Monthly and multi-year total terrestrial water storage anomalies derived from the GLOBGM and PCR-GLOBWB model, however, compared favorably with observations from the GRACE satellite. For the next versions of the GLOBGM further improvements require a more detailed (hydro)geological schematization and

better information on the locations, depths, and pumping rates of abstraction wells.

**1 Introduction**

The PCRaster Global Water Balance model (PCR-GLOBWB; van Beek et al., 2011) is a grid-based global-scale hydrology and water resource model for the terrestrial part of the hydrologic cycle, being developed at the Utrecht University, The Netherlands. This model, using a Cartesian (regular) grid representation for the geographic coordinate system (latitude and longitude), covers all continents except Greenland and Antarctica. For more than a decade, it has been applied to many water-related global change assessments providing estimates and future projections, e.g. regarding drought and groundwater depletion due to non-renewable groundwater withdrawal (Wada et al., 2010; de Graaf et al., 2017).

The latest version of PCR-GLOBWB (version 2.0; Sutanudjaja et al., 2018), or PCR-GLOBWB 2, has a spatial resolution of 5′ (5 arcminutes) corresponding to ~10 km the equator. It includes a 5′ global-scale groundwater model based on MODFLOW (Harbaugh, 2005), consisting of two model layers to account for confining, confined and unconfined aquifers (de Graaf et al., 2015, 2017). Recent publications have called for a better representation of groundwater in earth system models (Bierkens, 2015; Clark et al., 2015; Gleeson et al., 2021). Apart from providing a globally consistent and physically plausible representation of groundwater flow, global-scale groundwater models could serve to support global change assessments that depend on a global representation of groundwater resources. Examples of such assessments are the impact of climate change on vegetation, evaporation and atmospheric feedbacks (Anyah et al., 2008; Miguez-Macho and Fan, 2012), the role of groundwater depletion in securing global food security and trade (Dalin et al., 2017) and the contribution of terrestrial water storage change to regional sea-level trends (Karabil et al., 2021). From the work of de Graaf et al. (2017) and recent reviews (Condon et al. (2021) and Gleeson et al. (2021)), it follows that a number of steps is needed to make the necessary leap change in improving the current generation of global-scale groundwater models. Specifically these are: 1) improved hydrogeological schematization, particularly including multilayer semi-confined aquifer systems and the macroscale hydraulic properties of karst and fractured systems; 2) increased resolution to better resolve topography and in particular resolve smaller higher altitude groundwater bodies in mountain valleys; 3) improved knowledge on location, depth and rate of groundwater abstractions; 4) better estimates of groundwater recharge, especially in drylands and at mountain margins; 5) increased computational capabilities to be able to make simulations with the above mentioned improvements possible. Our paper specifically revolves around items 2) and 5) and focuses on the research question *If we improve spatial resolution, how should we make this computationally possible?* It is a small but necessary and important step to better global-scale groundwater simulation that needs to be taken to proceed further.

Here, we therefore consider a spatially refined version of the 5′ PCR-GLOBWB-MODFLOW global-scale groundwater model to 30″ resolution (~1 km at the equator), referred to as the GLOBGM in the following. The initial version of the GLOBGM, v1.0, that we developed in this study, has a maximum of two model layers and uses refined input from available PCR-GLOBWB simulations and the native parameterization of the global-scale groundwater model of de Graaf et al. (2017). Since we focus on the above-mentioned research question to improve spatial resolution, we note that improving schematization and parameterization is left for further research and for next versions of the GLOBGM.

Pushing forward to a 30″ resolution is a direct result for the growing availability of high-resolution datasets and the wish for exploiting the benefits of high performance computing (HPC) to maximize computer power and modeling capabilities (Wood et al., 2011; Bierkens et al., 2015). Typically, high-resolution Digital Elevation Maps (DEMs) are available derived from NASA Shuttle Radar Topography Mission products, such as HydroSHEDS (Lehner et al., 2008) at 30″ or the MERIT DEM (Yamazaki et al., 2017) at even 3″. Furthermore, subsurface data are becoming available in higher resolution, such as gridded soil properties at 250 m resolution from SoilGrids (Hengl et al., 2014), and global lithologies for ~1.8 million polygons from GLHYMPS (Huscroft et al., 2018). Although Moore's law still holds, viz. stating that processor performance doubles every two years, more and more processor cores are being added to increase performance. This means that making software

suitable for HPC inevitably requires efficient use of the available hardware. Anticipating this, we parallelized the groundwater solver for the GLOBGM as well as the pre-processing.

In this paper, the focus is on the technical challenges of implementing the GLOBGM using HPC given a performance target. The focus is not on improving schematization and/or parametrization and we use available PCR-GLOBWB data only. Furthermore, this study is mainly on transient simulation, since transient simulations make most sense for evaluating the effects of climate change and human interventions (see e.g. Minderhoud et al., 2017). We apply a similar approach as de Graaf et al. (2017) to obtain initial conditions for the GLOBGM: we use a steady-state result (under natural conditions; no pumping) to spin-up the model by running the first year back-to-back for 20 years to reach dynamic equilibrium. We restrict ourselves to present parallel performance results for transient runtimes only, since we found that steady-state runtimes with the GLOBGM are negligible compared to transient runtimes. We present a parallelization methodology for the GLOBGM and illustrate this by a transient experiment on the Snellius Dutch national supercomputer (SURFsara, 2021) for simulating 1958-2015 (58 years), including a 20-year spin-up. We provide a limited evaluation of the computed results, and we note that the current model is a first version that should be further improved in the future. We refer to Gleeson et al. (2021) for an extensive discussion on pathways to further evaluate and improve global-scale groundwater models. For this, the steady-state and transient results with the GLOBGM are compared to hydraulic head observations for the contiguous United States (CONUS), also considering the 5′ PCR-GLOBWB global-scale groundwater model, the 30″ global-scale inverse model of Fan et al. (2017), and the 250 m groundwater model for the CONUS of Zell and Sanford (2020). Furthermore, we compare simulated monthly and multi-year total water storage anomalies calculated with the GLOBGM and PCR-GLOBWB model, with observations from the Gravity Recovery and Climate Experiment satellite (GRACE) and its follow-on mission (Wiese, 2015; Wiese et al., 2016).

We use the model code program MODFLOW (Langevin et al., 2021, 2017), the most widely used groundwater modeling program in the world, being developed by the U.S. Geological Survey (USGS). The latest version 6 of MODFLOW supports a multi-model functionality that enable users to set up a model as a set of (spatially non-overlapping) submodels, where each submodel is tightly connected to other submodels at matrix level and has its own unique set of input and output files. In cooperation with the USGS, we parallelized this submodel functionality and created a prototype version (Verkaik et al., 2018, 2021c) that is publicly available and is planned to be part of a coming MODFLOW release. Building on our preceding research (Verkaik et al., 2021a, b), this prototype uses the message-passing interface (MPI; MPI Forum, 1994) to parallelize the conjugate gradient linear solver within the iterative model solution package (Hughes et al., 2017), supporting the additive Schwarz preconditioner as well as the additive coarse grid correction preconditioner (Smith et al., 1996).

This paper is organized as follows. In Section 2, first, the general parallelization approach for implementing the GLOBGM is given in Section 2.1, followed by the experimental set-up in Section 2.2, the workflow description in Section 2.3 and model evaluation methods in Section 2.4. In Section 3, the results are presented and discussed for the transient pre-processing in Section 3.1, the transient parallel performance in Section 3.2, the model evaluation in Section 3.3, and some examples of global-scale results are given in Section 3.4. Section 4 concludes this paper.

## 2 Methods

### 2.1 Parallelization approach

#### 2.1.1 General concept

The 5′ PCR-GLOBWB-MODFLOW global-scale groundwater model (GGM) consists of two model layers: where a
confining layer (having a lower permeability) is present, the upper model layer represents the confining layer and the lower
model layer a confined aquifer. If a confining layer is not present, both the upper and lower model layers are part of the same
unconfined aquifer (de Graaf et al., 2017, 2015). The GGM uses a structured Cartesian grid (geographic projection),
representing latitude and longitude, and includes all land and sea cells at the global 5′ extent. Using such grid means that we
have to take account for the fact that cell areas and volumes do vary in space, and therefore MODFLOW input for the recharge
and the storage coefficient need be corrected for this (see Sutanudjaja et al. (2011) and de Graaf et al. (2015) for details). Each
of the two GGM layers has 9.3 million 5′ cells (4,320 columns times 2,160 rows), and therefore the GGM has a total of ~18.7
million 5′ cells. A straightforward refinement of this grid to 30″ resolution would result in ~100 times more cells, hence 1.87
billion cells (two model layers of 43,200 columns times 21,600 rows). Creating and using such a model would heavily stress
runtime, memory usage and data storage.

For addressing this problem, we can significantly reduce the number of grid cells by applying unstructured grids and
maximize parallelism by deriving as many independent groundwater models as possible while satisfying all necessary
boundary conditions. This concept is illustrated by Figure 1. Starting with the 30″ global-scale land-sea mask and boundary
conditions prescribed by the GGM, we first derive independent unstructured grids and group them in a convenient way from
large to small (see Section 2.1.2 for details). Then, we define the GLOBGM as a set of four independent groundwater models:
three continental-scale groundwater models for the three largest unstructured grids and one remainder model called "Island
model" for the remainder of the smaller unstructured grids (see Section 2.1.3 for details). The unstructured grids for these
defined models are subject to parallelization: two partitioning methods (or domain decomposition methods) are considered
(see Section 2.1.4 for details): one for partitioning grid cells straightforwardly (grey arrows in Figure 1) and the other for
partitioning water catchments (red arrows in Figure 1). For each groundwater model, the chosen partitioning results in non-
overlapping subgrids that define the computational cells for the non-overlapping groundwater submodels, where the
computational work for each submodel is uniquely assigned to a processor core (MPI rank). Note that we deliberately reserve
the term "submodel" and "subgrid" for parallel computing. It should also be noted that deriving independent groundwater
models as we do in our approach could in principle also be executed for structured grids. However, this would introduce a
severe overhead of redundant cells for each independent groundwater model. Furthermore, in that case, defining the Islands
model would virtually make no sense since islands are scattered globally and the structured grid for this model would be
therefore almost as large as for the entire model.

In this work we focus on solving the groundwater models of the GLOBGM using distributed memory parallel computing
(Rünger and Rauber, 2013). We restrict ourselves to parallelization on mainstream cluster computers since they are often
accessible to geohydrological modelers. Cluster computers typically have distributed memory (computer) nodes, where each
node consists of several multicore-CPUs (sockets) sharing memory, and each node is tightly connected to other nodes by a
fast (high-bandwidth and low-latency) interconnection network. Instead of focusing on parallel speedups and scalability, which
are commonly used and valuable metrics for benchmarking parallel codes (e.g. Burstedde et al., 2018), we focus on a metric
that we believe is meaningful to the typical user for evaluating transient groundwater simulations: simulated-years-per-day
(SYPD). This metric, simply the number of years that a model can be simulated in a single day of 24 hours, has proven to be
useful for evaluating massively parallel performance in the field of atmospheric community modeling (Zhang et al., 2020).

Choosing a target performance $R_{tgt}$ in SYPD, we conduct a strong scaling experiment to estimate the number of cores and nodes to meet this target. For this, we select a representative but convenient groundwater model of the GLOBGM and determine the number of nodes to meet $R_{tgt}$ for a pre-determined maximum number of cores per node and a short period of simulation. Then, the target submodel size (grid cell count) is determined and used straightforwardly to derive the number of cores and nodes for all the groundwater models of the GLOBGM (see Section 2.1.5. for details). In the most ideal situation, using these estimates for the number of cores and nodes would result in a parallel performance that meets the target performance $R_{tgt}$ for each of the four groundwater models of the GLOBGM.

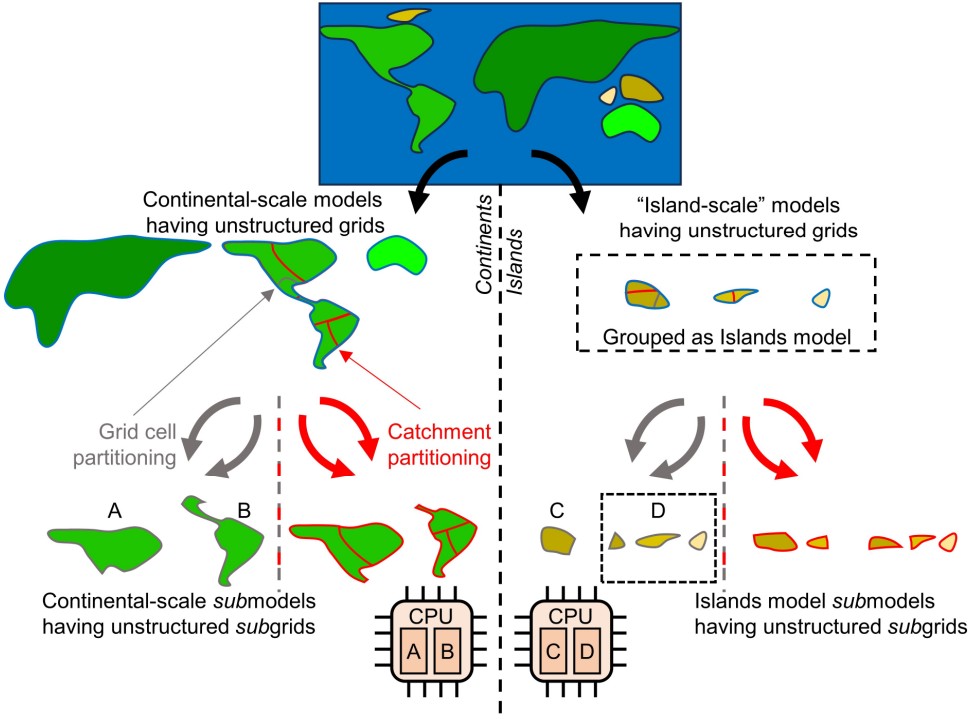

**Figure 1.** General concept of defining the GLOBGM as a set of independent (larger) continental-scale groundwater models (left in figure) and (smaller) "island-scale" groundwater models as a remainder model (right in figure). All these models have independent unstructured grids that are partitioned by a) straightforwardly assigning grid cells (grid cell partitioning; grey arrows) or b) assigning catchments (catchment partitioning; red arrows). The resulting subgrids define the computational cells for the submodels that are uniquely assigned to a processor core / MPI process. In this figure, this is illustrated by assigning submodels "A", "B" to one dual-core CPU and submodels "C" and "D" to another.

### 2.1.2    Procedure for deriving the independent unstructured grids

Starting point of the unstructured grid generation is a 30″ global-scale land-sea mask. Within the GGM, continents and islands are numerically separated (no implicit connection) by the sea cells and more precisely by a lateral Dirichlet boundary condition (BC) of a hydraulic head of 0 m for land cells near the coastline. Since this BC is only required near the coastline, this means that ~77% of the 5′ grid cells corresponding to sea are redundant. This motivates for applying unstructured grids for grid cell reduction. Using the 30″ mask, first, a minimal number of sea cells are added to the land cells in the lateral direction to provide for the Dirichlet BC by applying an extrapolation using a 5-point stencil. Second, from this resulting global map, all independent and disjoint continents/islands are determined, each having a (augmented) land-sea mask. Using this land mask, unstructured grid generation is done to account for the absence of the confining layer, where the GGM uses zero thickness (dummy) cells for the upper layer as a work-around for specifying a lateral homogeneous Neumann BC (no-flow). This holds for ~35% of the land cells of the GGM, also motivating the usage of unstructured grids for grid cell reduction. This means that in the GLOBGM two subsurface configurations are used: 1. Upper confining layer + lower confined aquifer (2 vertical cells); 2. (Lower) unconfined aquifer (1 vertical cell). For sake of convenience, we sometimes use "upper model layer" and "confining

layer" interchangeably, as well as "lower model layer" and "aquifer". Note that in the GLOBGM interaction with surface water
or surface drainage is modelled by putting rivers and drains in the first active layer, seen from top to bottom.

The resulting grids in the GLOBGM are clearly unstructured since the number of cell neighbors is not constant for all grid
cells, and therefore the grid cell index cannot be computed directly: in the lateral direction, constant-head cells (Dirichlet; 0
m) near the coastal shore are not connected to any neighboring canceled sea cells, and in the vertical direction we cannot
distinguish between upper and lower model layer anymore due to canceling of non-existing upper confining layer cells.
Because of this, we apply the Unstructured Discretization (DISU) package with MODFLOW (Langevin et al., 2017).

As a result of our procedure, we get $N_g = 9050$ independent unstructured grids, hence corresponding to 9050 independent
groundwater models, for total of ~278.3 million 30″ grid cells. Compared to the straightforward structured grid refinement of
the GGM to 30″ resolution, resulting in 1.87 billion grid cells (see Section 2.1.1), these unstructured grids give an 85% cell
reduction (land: 35% reduction, sea: 99.9% reduction). As a final step, we sort the unstructured grids by cell count, resulting
in $g_n$, $n = 1,\ldots,N_g$ grids, since this is convenient for identifying the largest and most computationally intensive groundwater
models subject to parallelization.

### 2.1.3    Defining the four groundwater models of the GLOBGM

Although from a modeling perspective, we might use all the 9050 derived independent groundwater models for the
GLOBGM, resulting from the procedure mentioned above (Section 2.1.2), we limit the number models for sake of coarse grain
parallelization and simplification of data management. Choosing a maximum of four ($N_m = 4$), the computational cells for
the three largest models are defined by unstructured grids $g_1$, $g_2$ and $g_3$, repetively: Afro-Eurasia (AE; 168 million 30″
cells), Americas (AM; 77 million 30″ cells) and Australia (AU; 16 million 30″ cells). The remaining 9047 smaller grids,
$g_4,\ldots,g_{9050}$, together for a total of 17 million 30″ cells, are grouped as "Islands" (ISL), see Figure 2. By doing this, we define
the GLOBGM as a set of four independent groundwater models that are subject to parallelization, of which three of them are
continental-scale groundwater models and one is the remainder model for the smaller islands (Islands model). The maximum
of four is motivated by the observation that the Islands model has almost the same number of cells as the Australia model, and
therefore we do not feel any need for using more groundwater models.

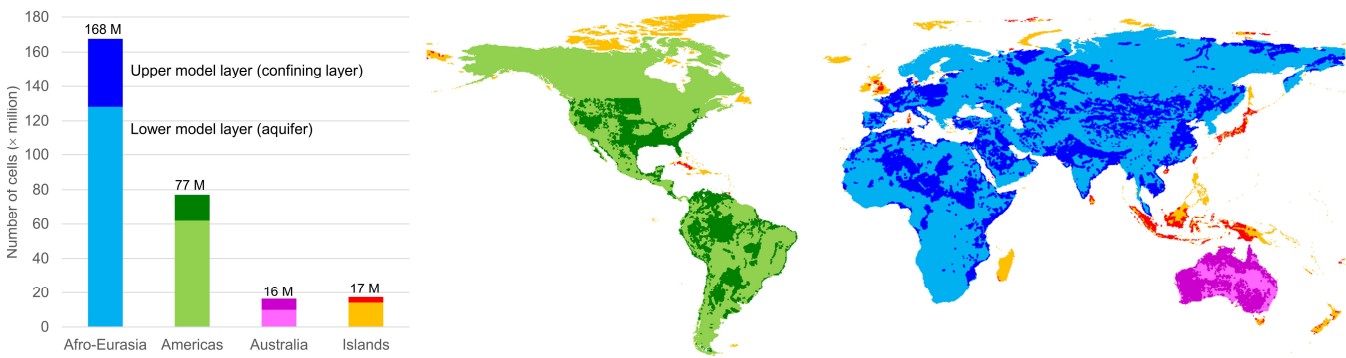

**Figure 2.** The four defined model areas of the GLOBGM having a total of 278.3 million cells. When the upper model layer is plotted this
means that there is also a lower model layer present. Otherwise, only the lower model layer is present.

### 2.1.4    Groundwater model partitioning: grid cell partitioning and catchment partitioning

For parallelizing the GLOBGM, each of the four groundwater models is partitioned into multiple non-overlapping
groundwater submodels, where each groundwater submodel is uniquely assigned to one processor core and one MPI process.
To obtain a good parallel performance, computational work associated with the submodels should be well balanced. For this,
we perform a grid partitioning using METIS (Karypis and Kumar, 1998), the most commonly used graph partitioner among
others like Chaco (Hendrickson and Leland, 1995) and Scotch (Pellegrini, 2008). Input for this partitioner is an undirected

graph, consisting of (weighted) vertices and edges, and the desired number of partitions. Here, we restrict our grid partitioning to lateral direction only, since the number of lateral cells is much larger than the number of model layers (here a maximum of two). This naturally minimizes the number of connections between the submodels and the associated point-to-point (inter-core) MPI communication times.

For each model $m^i$, $1 \le i \le N_{\mathrm{m}}$, a set of (coupled) submodels as a result of partitioning is defined by $m^{i,j}$, $j = 1, \ldots, N_{\mathrm{m}}^i$, where $N_{\mathrm{m}}^i$ is the number of submodels for model $i$. In our approach, each submodel is constructed by combining one or more areas $a_k^{i,j}$, $k = 1, \ldots, N_{\mathrm{a}}^{i,j}$, where $N_{\mathrm{a}}^{i,j}$ is the number of areas for model $i$ submodel $j$ of model $i$. Here, an area is defined as a 2D land surface represented by laterally connected, non-disjoint, $30''$ grid cells. The total number of areas for a model is defined by $N_{\mathrm{a}}^i = \sum_j N_{\mathrm{a}}^{i,j}$, and for the entire GLOBGM by $N_{\mathrm{a}} = \sum_i N_{\mathrm{a}}^i$. The general approach is to assign one or more areas to a submodel using METIS. Here after, we refer to this as area-based graph partitioning. For that, two partitioning strategies are considered, using:

1. METIS areas that are generated by straightforwardly applying METIS to model grids for $m^i$ to obtain $N_{\mathrm{m}}^i$ parts. We refer to this as the (straightforward) grid cell partitioning.
2. Catchment areas that are generated by rasterization and extrapolation of HydroBASINS catchments including lakes (Lehner, 2014). HydroBASINS catchments follow the Pfafstetter base-10 coding system for hydrologically coding river basins, where the main stem is defined as the path which drains the greatest area, and at each refinement level ten areas are defined: four major tributaries, five inter-basin regions and one closed drainage system (Verdin and Verdin, 1999). We refer to this as the catchment partitioning.

For straightforward grid cell partitioning we restrict ourselves to assigning exactly one METIS area to a submodel, hence $N_{\mathrm{a}}^{i,j} = 1$ for all $j$. In this approach, each lateral grid cell of model $m^i$ is subject to the METIS partitioner: each vertex in the graph corresponds to a lateral cell having a vertex weight equal to the number of vertical cells (1 or 2), and each edge corresponds to the inter-cell connection having an edge weight of 1 or 2, depending on the number of neighboring model layers. For catchment partitioning, typically $N_{\mathrm{a}}^{i,j} \gg 1$, each vertex in the METIS graph represents a catchment area: each vertex in the graph has a weight equal to the total number of grid cells within that catchment, and each edge corresponds to the inter-catchment connection having and edge weight equal to the sum of inter-cell connections to neighboring catchments.

Compared to straightforward grid cell partitioning, the graph being used by catchment partitioning is much smaller than the graph used by grid cell partitioning, since the number of catchments is generally much smaller than the number of lateral cells. Therefore, the METIS solver has much fewer degrees of freedom and is generally less optimal for catchment partitioning. However, catchment partitioning has several advantages compared to grid cell partitioning. First, the graphs are generally much smaller and therefore partitioning is significantly faster. In fact, this approach makes the grid partitioning almost independent on the grid cell resolution and is therefore suitable for even higher resolutions in the future. Second, catchment partitioning gives users the flexibility for giving the lateral submodel boundaries physical meaning. For example, choosing catchment boundaries simplifies a future parallel coupling to surface water routing modules (Vivoni et al., 2011). Third, this concept could be easily generalized for other types of areas, e.g., countries or states following administrative boundaries. Choosing such areas might simplify the management and maintenance of the submodels by different stakeholders within a community model.

Figure 3 illustrates the defined model entities for the case of one continent and three islands $(N_{\mathrm{g}} = 4)$, two models $(N_{\mathrm{m}} = 2)$ and seven catchments areas $(N_{\mathrm{a}} = 7)$. In this example, model $m^1$ has the largest (unstructured) grid $g_1$, and model $m^2$ the remaining smaller (unstructured) grids $g_2$, $g_3$ and $g_4$. The denoted cell numbers in Figure 3 correspond to the

numbers of model layers, where blue denotes a cell with a Dirichlet (sea) BC. In this example, each model has exactly two submodels (hence $N_m^1 = N_m^2 = 2$). For model $m^1$, catchments $a_1^{1,1}$ and $a_2^{1,1}$ are assigned to submodel $m^{1,1}$, and $a_1^{1,2}$ is assigned

to submodel $m^{1,2}$. For model $m^2$, catchment $a_1^{2,1}$ is assigned to submodel $m^{2,1}$, and $a_1^{2,2}$, $a_2^{2,2}$ and $a_3^{2,2}$ are assigned to submodel $m^{2,2}$. For this example, the load of model $m^1$ is well-balanced since each submodel has a load of 50. However, for model $m^2$, the first and second submodel have load 19 and 21, respectively, and therefore there is a load imbalance. Note that submodel $m^{2,2}$ has three disjoint subgrids, since in this case three islands are involved. This outlined situation for model $m^2$, where a submodel can have multiple disjoint subgrids, may occur in the global domain decomposition for the Islands model since grid cell partitioning is done for all 9047 disjoint grids together. In general, the presence of disjoint subgrids may occur for any submodel within the GLOBGM, also for the Afro-Eurasia, Americas, and Australia models. The reason is that none of the METIS solvers can guarantee contiguous partitions. However, we here use the multilevel recursive bisection option, that is known to give best results in that respect.

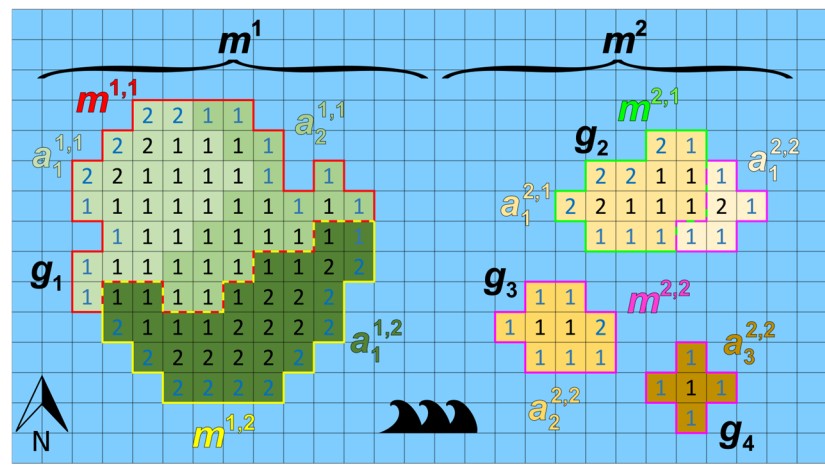

**Figure 3.** Example of an imaginary world containing one large continent in the West (having unstructured grid $g_1$) and three smaller islands in the East (with unstructured grids $g_2$, $g_3$, and $g_4$ from smallest to largest). The world's model is chosen to be a set of two models: one continental-scale model $m^1$ ($g_1$) and one "islands model" $m^2$ ($g_2$, $g_3$, and $g_4$). In this example, each of the model consists of exactly two submodels that are determined using catchment partitioning (strategy 2): for model $m^1$, catchments $a_1^{1,1}$ and $a_2^{1,1}$ are assigned to submodel $m^{1,1}$, and $a_1^{1,2}$ to submodel $m^{1,2}$. For model $m^2$, catchments $a_1^{2,1}$ and $a_1^{2,2}$ are assigned to submodel $m^{2,1}$, and $a_2^{2,2}$ and $a_3^{2,2}$ to submodel $m^{2,2}$. Note that in this example model $m^2$ is the remainder model and is chosen to consists of three islands, that each could have been chosen as a separate independent model. Furthemore, note that for this model $m^2$ submodel $m^{2,2}$ consists of three disjoint subgrids. The numbered cells denote the cells included in the model: the number equals the number of vertical cells; blue numbers denote constant head (Dirichlet) cells with a head of 0 m.

### 2.1.5 Node selection procedure

When evaluating the target performance $R_{tgt}$, we restrict ourselves to model runtime only. Since pre- and post-processing are very user specific, associated runtimes cannot be generalized. However, we do consider them in our analysis. Processing runtimes can be substantial, making parallel processing inevitable, as we will illustrate for our limited pre-processing. To keep pre-processing runtimes feasible, we apply embarrassing parallelization (see e.g. Herlihy et al., 2020) where processing for input data (tiles and submodels) is done independently using multiple threads simultaneously on multiple nodes.

For achieving $R_{tgt}$, we need a selection procedure for estimating the number of (processor) cores for the three continental-scale groundwater models and the Islands model. Since it is common to take the number of cores per node at a fixed value ($N_{CPN}$), this means that choosing the number of cores is equivalent to choosing the number of nodes. Ideally, we would take $N_{CPN}$ to be equal to the total number of cores within a node and therefore maximize computer resources utilization. However,

it is not always advantageous or even impossible to use the total number of cores due to the memory access constraint. In practice, the best performance is often obtained using a lesser number of cores. For example, the linear conjugate gradient solver that is used for solving the groundwater flow equation, dominates runtime and is strongly memory-bound because of the required sparse matrix-vector multiplications (Gropp et al., 1999). This means that, starting from a certain number of cores, competition for memory bandwidth hampers parallel performance and contention is likely to occur (Tudor et al., 2011). Typically, this directly relates to the available memory channels within a multicore-CPU, linking the RAM and processor cores. In our approach, we first determine $N_{CPN}$ by performing a strong scaling experiment (keeping the problem size fixed) within a single node for the well-known high performance conjugate gradients supercomputing benchmark (HPCG; Dongarra et al., 2016). By doing this, we assume that HPCG is representative for the computation in MODFLOW. The performance metric for this experiment is floating-point operations per second (FLOPS), that is commonly used for quantifying numerical computing performance and processor speed. Then, by conducting a strong scaling experiment for a single (medium-sized) model and a short simulation period while keeping $N_{CPN}$ fixed, the number of nodes is selected such that $R_{tgt}$ is achieved. This gives the preferred submodel size that is used to determine the number of nodes for all other models; see Section 2.3.1 for more details.

## 2.2 Experimental set-up

### 2.2.1    Description of the GLOBGM and application range

Table 1 and Table 2 show the datasets used to parameterize the GLOBGM in this study. For details on the MODFLOW model description and conceptualization, we refer to the preceding (de Graaf et al., 2019, 2017, 2015; Sutanudjaja et al., 2014, 2011). The application domain of the GLOBGM is similar to that of the 5′ PCR-GLOBWB global-scale groundwater model (GGM; de Graaf et al., 2017). We therefore note that the limitations of the GGM also apply to the GLOBGM. For clarity we repeat these limitations here: 1) the GGM is intended to simulate hydraulic heads in the top aquifer systems, so unconfined aquifers and the uppermost confining aquifers; 2) wherever there are multiple stacked aquifer systems, these are simplified in the model to one confining layer and one confined aquifer; 3) the model schematization is suitable for hydraulic heads in large sedimentary alluvial basins (main productive aquifers) that have been mapped at a 5′ resolution; 4) in as far these sedimentary basins include karst, it is questionable if a Darcy approach can be used to simulate large-scale head distributions; 5) due to the limited resolution of the hydrogeological schematizations, in mountain areas we simulate the hydraulic heads in the mountain blocks but not those of groundwater bodies in hillslopes and smaller alluvial mountain valleys; 6) also, for the hydraulic heads in the mountain blocks, we assume that secondary permeability of fractured hard rock can also simulated with Darcy groundwater flow; an assumption that may be questioned.

In this study, we adopted an offline coupling approach that used the PCR-GLOBWB model output of Sutanudjaja et al. (2018) as the input to the MODFLOW groundwater model. The PCR-GLOBWB model output that was used consists of monthly fields for the period 1958-2015 for the variable groundwater recharge (i.e., net recharge obtained from deep percolation and capillary rise), groundwater abstraction, and runoff. The latter was translated to monthly surface water discharge by accumulating it through the 30″ river/drainage network of HydroSHEDS (Lehner et al., 2008). From the monthly surface water discharge fields, we then estimated surface water levels using Manning's equation (Manning, 1891) and the surface water geometry used in the PCR-GLOBWB model.

The groundwater model simulation conducted consists of two parts: a steady-state and a transient simulation. For both simulations, default solver settings from the 5′ model were taken for evaluating convergence, and within the linear solver we did not apply coarse grid correction. We started with a steady-state MODFLOW model using the average PCR-GLOBWB runoff and groundwater recharge as the input. No groundwater abstraction was assumed for the steady-state model therefore representing a naturalized condition, as the simulated steady-state hydraulic heads will be used as the initial conditions for the

transient simulation (i.e., assuming low pumping in ~1958). The transient MODFLOW simulation was calculated at daily time steps with monthly stress period input of surface water levels, groundwater recharge and groundwater abstraction. Using the steady-state estimate as the initial hydraulic heads, the model was spun-up using the year 1958 input for 20 years (to further

warm-up the initial states and add the small effect of 1958 pumping), before the actual transient simulation for the period 1958-2015 started.

**Table 1.** Input datasets used for the GLOBGM transient model parameters.

| Item | Number of grids (fixed / monthly) | Main sources/references, original spatial resolution/support | Remarks |
|---|---|---|---|
| Model upper layer top elevation, DEM (digital elevation model) | 1 / - | MERIT Hydro DEM (Yamazaki et al., 2017), 3" resolution | We upscaled the 3" MERIT DEM to 30" resolution, e.g., for the estimate of surface elevation. Yet, we also used its original 3" values to derive several other elevation values (at 30" resolution), such as the elevations of flood plain, river head, river bottom, and drainage bottom. For detailed descriptions, we refer to the preceding papers (de Graaf et al., 2019, 2017, 2015; Sutanudjaja et al., 2014, 2011). |
| Model layer thicknesses (confining layer and aquifer thicknesses) | 2 / - | de Graaf et al., 2017; Sutanudjaja et al., 2018, 5' resolution | We performed a bilinear interpolation to bring them to 30" resolution. |
| Hydraulic conductivities and storativities (storage coefficients) | 6 / - (2 for horizontal conductivities, 2 for vertical conductivities, and 2 for storativities) | GLHYMPS (Gleeson et al., 2014); GLiM (Hartmann and Moosdorf, 2012) a polygon map with the scale 1:3,750,000 | We rasterize the polygon map of GLiM (Hartmann and Moosdorf, 2012) to 30" resolution. The values assigned for each class are based on GLHYMPS (Gleeson et al., 2014). |
| Groundwater recharge | - / 1 | PCR-GLOBWB (Sutanudjaja et al., 2018), 5' resolution | We simply resampled/mapped this map to 30" (no downscaling) |
| Groundwater abstraction (pumping wells) | - / 2 | PCR-GLOBWB (Sutanudjaja et al., 2018), 5' resolution | We simply resampled/mapped this map to 30" (no downscaling) |
| River package | - / 3 | See Table 2 for the input for river and drainage packages. | See Table 2 for the input for river and drainage packages. |
| Drainage package | 1 / 2 | See Table 2 for the input for river and drainage packages. | See Table 2 for the input for river and drainage packages. |
| | 10 / 8 | | |

**Table 2.** Input datasets used for the MODFLOW river and drainage packages.

| Item | Main sources/references, original spatial resolution/support | Remarks |
|---|---|---|
| River/drainage network | HydroSHEDS (Lehner et al., 2008) | Used as the network to accumulate local runoff for estimating surface water discharge. |
| Local runoff | PCR-GLOBWB 2 (Sutanudjaja et al., 2018), 5' resolution | Used to derive surface water discharge and surface water level fields. |
| Surface water discharge and surface water level | This study, 30" resolution | Surface water discharge was estimated by accumulating runoff through the 30" river/drainage network of HydroSHEDS (Lehner et al., 2008). Then, surface water level was calculated from discharge using the Manning's equation (Manning, 1891). |
| Groundwater recession coefficient (used to estimate drainage conductance) | This study, 30" resolution | Calculated following the drainage theory of van de Leur Kraijenhoff (1958) based on the drainage network density and aquifer properties. For the drainage density, we used the estimate from van van Beek and Bierkens (2009). The aquifer properties were from Gleeson et al. (2014). |

### 2.2.2    Run-time target in simulated-years-per-day

As an experiment, we consider the typical "9 to 5" user, who likes to start a simulation at 5 PM and get the results next working day at 9 AM. The target for our transient experiment for 1958-2015 is then to simulate 78 years (58 + 20 years spin-up) in 16 hours. Accordingly, this is 0.67 days, and therefore we set $R_{tgt} = 78/0.67 = 117$ SYPD.

### 2.2.3    Dutch national supercomputer Snellius

All our experiments were conducted on the Dutch national supercomputer Snellius (SURFsara, 2021), a cluster computer
using multicore CPUs that is easily accessible to users from Dutch universities and Dutch research institutes. It consists of heterogeneous distributed memory nodes (servers), where we restrict to using default worker nodes having 256 GB[1] memory ("thin" nodes), see Table 3, that are tightly connected through a fast interconnection network having a low latency and high bandwidth. Each node houses two 64-core AMD CPUs that connect to 256 GB of local memory through two sockets. For this study, a storage of 50 TB and a maximum of 3.8 million files is used that is tightly connected to the nodes, enabling parallel
I/O using the Lustre parallel distributed file system. Since the Snellius supercomputer is a (more or less) mainstream cluster built with off-the-shelf hardware components, we believe that our parallelization approach is well applicable to many other supercomputers.

**Table 3.** Configuration of the default computing nodes used on the Dutch national supercomputer Snellius (SURFsara, 2021).

| | |
|---|---|
| Number of nodes | 504 |
| Number of sockets per node | 2 |
| Multicore-CPU per socket | AMD EPYC 7H12 (2nd gen. Rome) |
| | 64 cores, 2.6 GHz, 8 memory channels |
| Memory (RAM) per node | 256 GB, 3200 MHz, DDR4 (16 DIMMs) |
| Interconnection network | Fat tree topology, InfiniBand HDR100 |
| File system | Lustre parallel with striping, InfiniBand HDR100 |

## 2.3  Workflow description

### 2.3.1    Model Workflow and Node Selection Workflow

The main workflow for pre-processing and running the four models within the GLOBGM is given by Figure 4a. The so-called Model Workflow uses data that are initially generated by the process "Write Tiled Parameter Data", that writes the 30″ grids (see Table 1) using a tiled-based approach in an embarrassingly parallel way, see Section 2.3.2 for more details. For the
Model Workflow, prior to the actual partitioning and writing the model files in an embarrassingly parallel way (viz. the process "Partition & Write Model Input"; see Section 2.3.4), a preparation step is done for the partitioning (the process "Prepare Model Partitioning"; see Section 2.3.3). The general idea behind this preparation step is to simplify the embarrassingly parallelization for generating the submodels, by storing all required mappings associated to the unstructured grids for the continents and islands, boundary conditions, areas, connectivities, and data tiling. By doing this, we do not have to recompute the mappings
for each submodel and therefore save runtime. First, these mappings are used to assign the areas for the catchment partitioning, therefore requiring runtime that is negligible. Second, these mappings are used for fast directaccess read of (tiled parameter) data required for assembling the submodels in parallel. The final step in the Model Workflow is to run the models on a distributed memory parallel computer (process "Run Model" in Figure 4a).

---

[1] The abbreviations GB (gigabyte) and TB (terabyte) denote the size of the binary (base-2) memory system.

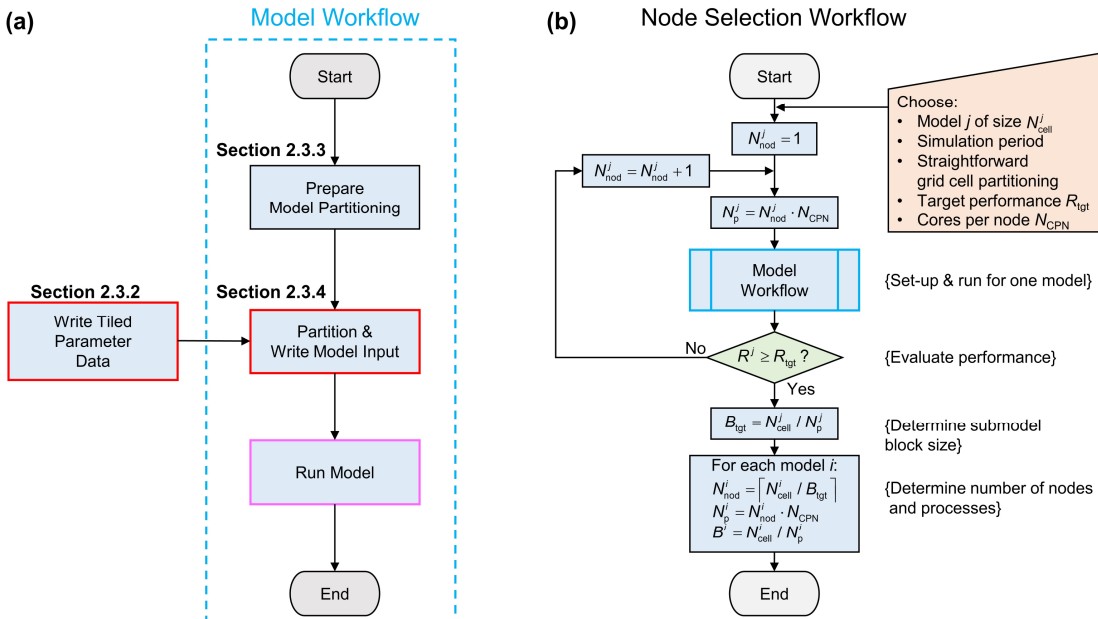

**Figure 4.** (a): Model Workflow for processing the GLOBGM. A red box and pink box denote embarrassingly parallel and distributed memory parallel, respectively, see Section 2.1. (b): Node Selection workflow using the Model Workflow. Workflow symbols used: process (blue), decision (green), manual input (orange).

It should be noted that in our workflow we have deliberately left out the post-processing used in this study. The reason for this is that we only perform limited post-processing and mainly focus on measuring the performance of model simulation. In our post-processing, runtimes and data storage for transient evaluation are low since we use direct-access reading of binary output and only generate time series for a limited number of selected well locations. However, in real-life application of the GLOBGM, more output data may be needing to be processed, resulting in non-neglectable runtimes. In that case, post-processing could benefit from parallelization.

Figure 4b shows the Node Selection workflow, which uses the Model Workflow (Figure 4a). The main purpose of the Node Selection Workflow is to estimate the number of nodes $N_{\mathrm{nod}}^{i}$ to be used for each model $i$. This estimation is done by conducting a strong scaling experiment, meaning that the number of cores is being varied for a model having a fixed problem size. Input for this workflow is a selected model $1 \leq j \leq N_{\mathrm{m}}$, having a convenient grid size ($N_{\mathrm{cell}}^{j}$) and simulation period. Straightforward grid cell partitioning is applied, see Section 2.1.4, as well as a straightforward iteration scheme. Starting from $N_{\mathrm{nod}}^{j} = 1$, in each iteration of this workflow, the number of cores (or MPI processes) $N_{\mathrm{p}}^{j}$ is chosen to be a multitude of the number of cores per node $N_{\mathrm{CPN}}$, hence $N_{\mathrm{p}}^{j} = N_{\mathrm{nod}}^{j} \cdot N_{\mathrm{CPN}}$. Then, the Model Workflow generates the model input files for this number of cores, followed by running the model to obtain runtime performance $R^{j}$. The iteration finishes when $R^{j} \geq R_{\mathrm{tgt}}$, and the target submodel size (number of grid cells) is determined by $B_{\mathrm{tgt}} = N_{\mathrm{cell}}^{j} / N_{\mathrm{p}}^{j}$. Using this submodel size, for each model $i = 1, \ldots, N_{\mathrm{m}}$, the number of nodes is given by $N_{\mathrm{nod}}^{i} = \lceil N_{\mathrm{cell}}^{i} / B_{\mathrm{tgt}} \rceil$, where $\lceil \cdot \rceil$ denotes the ceiling function. Then, the number of cores to be used for each model $i = 1, \ldots, N_{\mathrm{m}}$ follows straightforwardly from $N_{\mathrm{p}}^{i} = N_{\mathrm{nod}}^{i} \cdot N_{\mathrm{CPN}}$. The (maximum) total number of nodes being used for the GLOBGM we denote by $N_{\mathrm{nod}} = \sum_{i} N_{\mathrm{nod}}^{i}$.

### 2.3.2 Model Workflow: Write Tiled Parameter Data

In the offline coupling of PCR-GLOBWB to MODFLOW (see Section 2.2.1), all model parameter data required for the MODFLOW models are written prior to simulation. For this, the PCR-GLOBWB-MODFLOW Python scripts that use PCRaster Python modules (Karssenberg et al., 2010) are slightly modified for the MODFLOW module to process and write







30″ PCRaster raster files (uncompressed, 4-bytes single-precision). The processing is done for a total of 163 squared raster tiles of 15° having 30″ resolution ($1800 \times 1800$ cells), enclosing the global computational grid, see Figure 5. Using tiles has benefits for two reasons. First, using tiles cancels a significant number of redundant sea cells (missing values), reducing the data storage for storing one global map from 3.47 GB to 1.97 GB in our case (43% reduction). Second, using tiles allows for (embarrassingly) parallel pre-processing to reduce runtimes. In our pre-processing, the 163 tiles are distributed proportionately over the available $N_{nod}$ nodes. It should be noted that, although the tiling approach is quite effective, still many coastal tiles have redundant sea cells. Since missing values do not require pre-processing, one might argue that these tiles would result in a parallel workflow imbalance. However, the version 4.3.2 of PCRaster we used does not treat missing values differently, and therefore choosing tiles of equal sizes seems to be appropriate. It should also be mentioned that choosing 15° tiles is arbitrary and different tile sizes could be chosen.

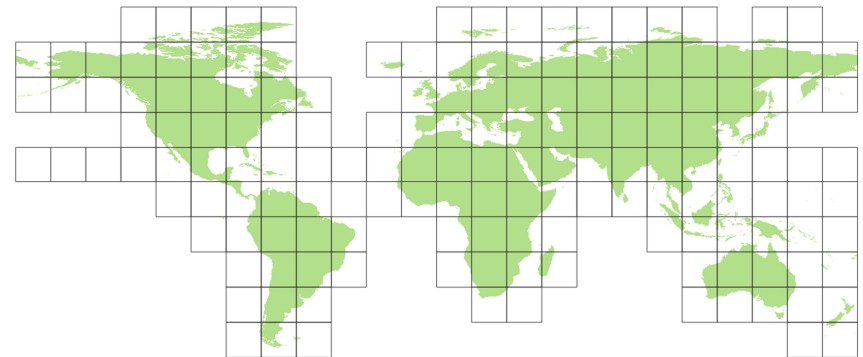

**Figure 5.** Parameter data tiles having 30″ resolution and 15° size ($1800 \times 1800$ cells). Total: 163.

### 2.3.3 Model Workflow: Prepare Model Partitioning

The workflow for preparing the partitioning is given by Figure 6 in more detail. This workflow derives and writes all necessary mappings that are being used for the process "Partitioning & Write Model Input", see Section 2.3.4. It includes the two options for generating areas, see Section 2.1: METIS areas and catchment areas, see processes "A" and "B" in Figure 6, respectively.

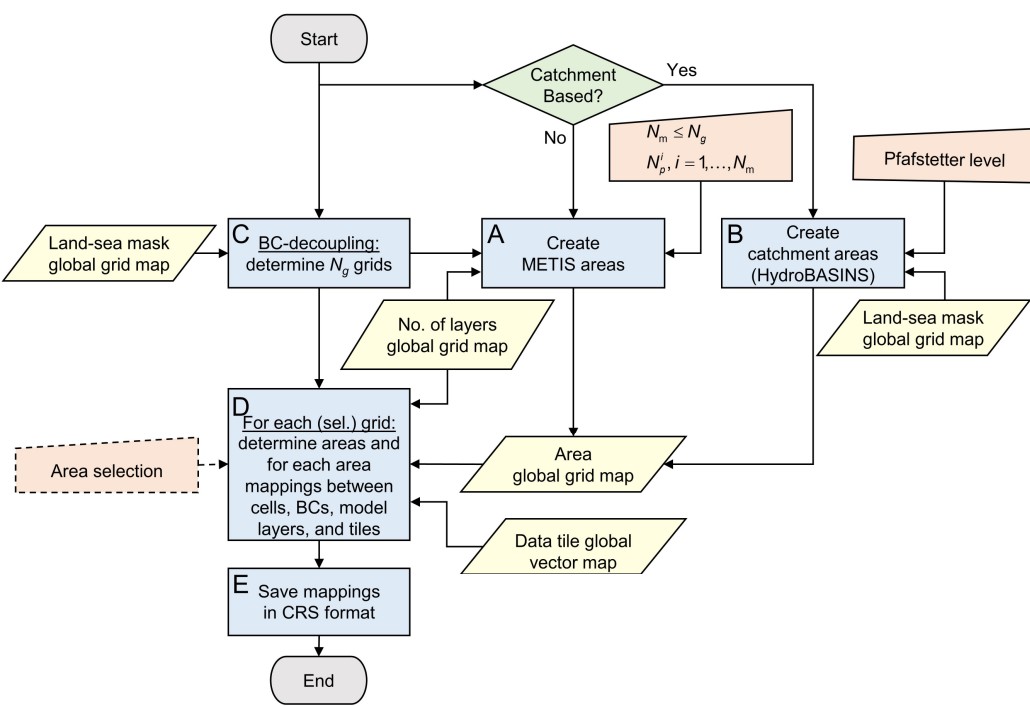

**Figure 6.** Process "Prepare Model Partition" in detail of the workflow in Figure 4a. Abbreviations: BC = Boundary Condition; CRS = Compressed Row Storage. Workflow symbols used: process (blue), decision (green), manual input (orange), input data (yellow).

When METIS areas are chosen to prepare for straightforward grid cell partitioning, areas are derived from grids $g_n$, $n = 1, \ldots, N_g$, that are disjoint and increasing in size corresponding to continents and islands, see Section 2.1 and process "C" in Figure 6. For each (continental-scale) model $i$, $1 \leq i < N_m$, exactly one grid $g_i$ is taken, and for the remainder (islands) model $i = N_m$, the grids $g_n$, $n = N_m, \ldots, N_g$ are taken. Applying METIS partitioning for these grids to obtain $N_p^i$ partitions

is then straightforward, see Section 2.1.4.

When catchment areas are chosen to prepare for catchment partitioning, HydroBASINS catchments are determined at the global 30″ extent, see process "B" in Figure 6. This is done for a given Pfafstetter level that directly relates to the number of catchments. First, local catchment identifiers (IDs) for the HydroBASINS polygons (v1.c including lakes) are rasterized to 30″ for the corresponding eight HydroBASINS regions, numbered consecutively, and merged to the global 30″ extent. Second,

extrapolation of the catchment IDs in lateral direction near the coastal zone is done using the (extended) global 30″ land-sea mask and a 9-point stencil operator. Third, the 9-point stencil operator is used to identify independent catchments and new IDs are generated and assigned where necessary.

The result of creating METIS or catchment areas is stored in a global 30″ map with unique area IDs. This map is input for process "D" in Figure 6, together with the grid definition, a (vector-based) definition of the data tiles with bounding boxes,

and a global 30″ map with the number of model layers per cell. Optionally, areas can be selected by a user defined bounding box, by polygon or ID, allowing users to prepare a model just for a specific area of interest. However, in this study we limit ourselves to the global extent only and therefore this option is not being used. In process "D", for each continent or island, all required mappings are determined for each of the covering areas: global index cells numbers and bounding box, neighboring areas and cell-to-cell interfaces, location of boundary conditions, the connected data tiles, and number of model layers per cell.

An undirected graph is constructed using the compressed row storage (CRS) format for efficient data storage with pointers (counters) to the (contiguous) bulk data with mappings. In process "E" in Figure 6, these mappings are saved to binary files that allows for fast direct access reading of all mappings required for the partitioning and writing the model input (see process "Partitioning & Write Model Input", Section 2.3.4).

### 2.3.4 Model Workflow: Partition & Write Model Input

The workflow for the partitioning and writing the model input files (Figure 7) consists of three processes: the "Area-Based Graph Partitioning" (A), "Assemble & Write Submodel Input Data" (B), and "Assemble & Write Inter-Submodel Connections" (C). This workflow is set up in a flexible way such that it enables embarrassingly parallel computing for a given range of models and submodels. This means that for selected models and submodels, data can be processed independently by using fast area-based graph partitioning and using the pre-defined and stored mappings.

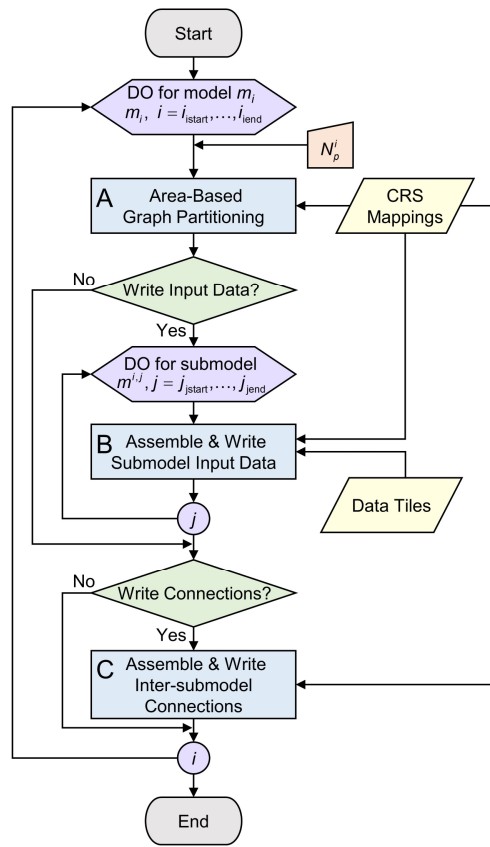

**Figure 7.** Process "Partition & Write Model" in detail of the Model Workflow in Figure 4a. Workflow symbols used: process (blue), decision (green), manual input (orange), input data (yellow), iterator (purple).

The first step is to perform the partitioning, see "A" in Figure 7. The goal is to determine the submodel partitions by optimally assigning areas to balance the parallel workload using METIS while minimizing the edge-cuts. For a model $m^i$, first all necessary area data (cumulative vertex and edge weights, connectivities) are collected from the saved mappings that are a result of the "Prepare Model Partitioning" workflow, see 2.3.3. For each model $m^i$, $i = 1,\ldots,N_{\mathrm{m}} - 1$, this is done for all areas belonging to grid $g_i$; for the remainder model $m^{N_{\mathrm{m}}}$ this is done for all the areas belonging to grids $g_{N_g},\ldots,g_{Ng}$. After constructing the graph for model $m^i$, METIS is being called to partition the graph into user-defined $N_{\mathrm{p}}^i$ parts.

Figure 8 depicts the graphs associated with the example of Figure 3. The vertices correspond to catchments and the edges to the connectivities between the catchments. Since in this example the number of disjoint grids $N_{\mathrm{g}} > N_{\mathrm{m}}$, which is generally the case, the associated graph for the (remainder) model $m^2$ is disconnected, while the graph for model $m^1$ is connected. The vertex weights are defined as the sum of the 30″ cell weights for a corresponding catchment, and the edges are defined as the sum of shared cell faces between the catchments, see Section 2.1. Focusing on the vertex weights for the example graph, let $W^i$ denote the model weight of model $i$ and $w^{i,j}$ the submodel weight such that $W^i = \sum_j w^{i,j}$. In this example, model $m^1$ has a total weight of $W^1 = 100$ divided into two submodels, each having a weight of $w^{1,1} = w^{1,2} = 50$. Following Karypis and Kumar (1998), we here define the load imbalance for a model $i$ as the maximum submodel weight divided by the average (target) weight, hence $I_i = \max_j w^{i,j} / (W^i / N_p^i) = N_p^i \cdot \max_j w^{i,j} / W^i$. In this definition, the model is perfectly balanced when $I_i = 1$ and load imbalance occurs when $I_i > 1$. Assuming Amdahl's law holds (Amdahl, 1967), it follows that the speedup (and hence parallel performance) is proportionally to $I_i^{-1}$. Defining the imbalance increase as $I_i^* = 100 \cdot (I_i - 1)$ [%], then for model $m^1$ the submodels are perfectly balanced ($I_1^* = 0$ %). On the other hand, the catchments for model $m^2$ could not be

perfectly distributed over the two submodels since the second submodel has a larger weight, $w^{2,2} = 21$, than the first submodel, $w^{2,1} = 19$. Therefore, the load imbalance for the second model, having a total weight $W^2 = 40$, is $I_2^* = 5$ %.

In general, area-based partitioning with catchments results in an insurmountable load imbalance, depending on several factors. The amount of load imbalance depends on the (multi-level) partitioning algorithm being used, the number of
catchments related to the number of partitions, the catchment geometry, and the effect this all has on the search space. In this study, we do not try to make any quantitative or qualitative statements on this. Moreover, we aim for the practical aspects for a given commonly used graph partitioner with default settings applied to a realistic set of catchments. It should be noted, as already highlighted in Section 2.1, that load imbalance for METIS areas can be considered as optimal, since the number of vertices (equals the number of lateral cells) is very large compared to the number of partitions, and do not vary strongly in
weight. It is verified that the obtained load imbalance never exceeds the specified maximum tolerance of $I_i = 1.0001$.

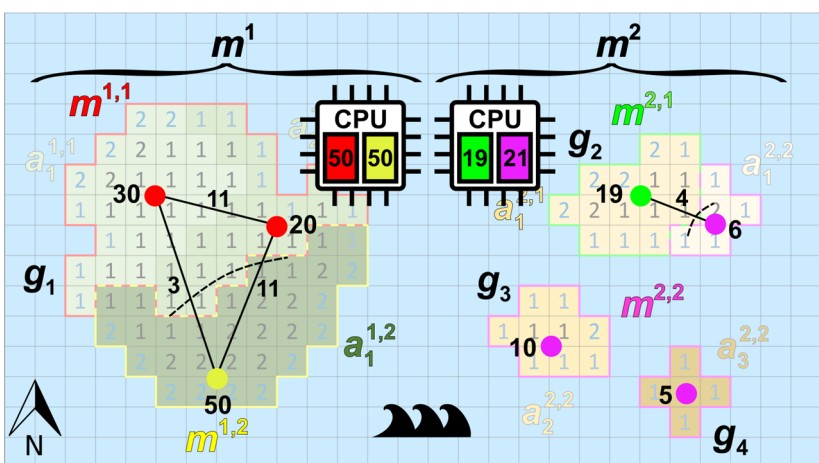

**Figure 8.** Example of area-based graph partitioning corresponding to the example of Figure 3, where the work for each submodel is uniquely assigned to a core of a dual-core CPU.

The second step in the workflow of in Figure 7, see "B", is to process all submodel data. First, the resulting METIS or catchment areas assigned by METIS (result of process "A") are used to assemble the unstructured grid. Then, using the tiled parameter data and the CRS mappings, both steady-state as well as transient model input data are written. To significantly reduce the number of model input files for transient simulation, each submodel has exactly one binary file for storing all necessary bulk data. This is a new functionality of our MODFLOW 6 prototype. For the third and last step in the workflow of
Figure 7, process "C", first an inter-submodel graph is assembled for each model by the merging the inter-area graph from the CRS mappings and the derived partitions from process "A". Second, the submodel connections (MODFLOW exchanges) are written to files, as well as wrappers for running the steady-state model, the transient spin-up model, and the actual transient model.

## 2.4  A limited first evaluation of the GLOBGM

Here, the GLOBGM is evaluated for the CONUS for which many head measurement data sets are publicly available. For our study, we restrict to selected head observations from the USGS National Water Information System (NWIS) database (USGS, 2021). For the global evaluation, where transient head time series are limited, we evaluated groundwater level fluctuations with total water storage anomalies derived from the GRACE and GRACE-FO satellites (Wiese, 2015; Wiese et al., 2016). This evaluation is limited, however, since the GLOBGM v1.0 is an initial model. This means various other aspects
are left for further research: improving model schematization (e.g., geology), improving model parameters by adding more (regional scale) data and calibration, and adding more global dataset for comparison. Our evaluation is mainly on comparison to the 5′ PCR-GLOBWB-MODFLOW model (GGM) for which we used consistent upscaled data.

### 2.4.1    Steady-state hydraulic heads for the CONUS

The steady-state (i.e., long-term average) evaluation is limited to the so-called natural condition, meaning that human intervention by groundwater pumping is excluded from the model. Besides comparing the GLOBGM to the GGM (of de Graaf et al., 2017), we compare the computed steady-state hydraulic heads to two other models: the global-scale inverse model (GIM) from Fan et al., (2017), having 30″ resolution and used for estimating steady-state root water-uptake depth based on observed productivity and atmosphere by inverse modeling, and the CONUS groundwater model from Zell and Sanford (2020), a continental-scale MODFLOW 6 groundwater model developed by the USGS for simulating the steady-state surficial groundwater system for the CONUS. From here, these latter models are referred to as GIM and CGM, respectively. It should be noted that contrary to the GLOBGM and the GGM, the GIM and the CGM have been calibrated on head observations. For the steady-state evaluation, we use the same NWIS wells as selected by Zell and Sanford (2020) to evaluate the performance of the four models. Mean hydraulic head residuals are computed for HUC4 surface water boundaries from the USGS Watershed Boundary Dataset (U.S. Geological Survey, 2021b) to get a spatially weighted distribution of the residuals for the CONUS.

### 2.4.2    Transient hydraulic heads for the CONUS

For the simulation period 1958-2015, GLOBGM computed hydraulic heads at monthly time step are compared to NWIS time-series considering the non-natural condition (including groundwater pumping). We perform the same evaluation for a recent GGM run (de Graaf et al., 2017) and compare the outcome with the GLOBGM to evaluate a possible improvement. Closely following the methodology as chosen in de Graaf et al. (2017) and Sutanudjaja et al. (2011), we compute the following long-term averaged statistics: the sample correlation coefficient $r_{mo}$ is used for quantifying the timing error between model ("m") and observation ("o"). Furthermore, the absolute and relative interquartile range error, $\text{IQRE}_{mo} = |\text{IQR}_m - \text{IQR}_o|/\text{IQR}_o$, with $\text{IQR}_m$ and $\text{IQR}_o$ the interquartile ranges for the model and observations, respectively, is used to quantify the amplitude error. Additionally, the trend of the (monthly averaged) time series is computed, considering the slope $\beta_y$ of yearly averaged hydraulic heads from a simple linear regression with time. We assume that a time series has a trend if $|\beta_y| > 0.05$ m per year, and no trend otherwise.

Transient evaluation using measurements from the NWIS database requires filtering out incomplete time series and data locations for areas that are not represented by the GLOBGM. Similar to de Graaf et al. (2017), time-series are selected having a record covering at least five years and include seasonal variation. However, different from de Graaf et al. (2017), we only consider well locations for sedimentary basins (including karst). Furthermore, we aggregate the computed statistics (timing, amplitude, trend) to the same HUC4 surface water boundaries as used for the steady-state evaluation (see Section 2.4.1) for obtaining results at a scale that is commensurate with a global-scale groundwater model. In this, we declare HUC4s having lesser than five NWIS wells not spatially representative and exclude them from the presented statistics. In total, the filtering resulted in 12,342 site locations selected for comparison between simulated and observed head time series (see also the supplementary information).

### 2.4.3 Total water storage anomalies for the world's major aquifers

For the transient evaluation, we also computed the simulated total water storage (TWS) and compared it to the one estimated from GRACE gravity anomalies. For the simulated TWS, we used computed hydraulic heads from the GLOBGM (multiplied times the storage coefficient) and data (for snow, interception, surface water, and soil moisture) from the PCR-GLOBWB run of of Sutanudjaja et al. (2018). Here we compared the monthly simulated TWS to the monthly gravity solutions from GRACE and GRACE-FO as determined from the JPL RL06.1Mv03CRI (Wiese, 2015; Wiese et al., 2016). For this evaluation, cross-correlations between simulated and observed TWS anomalies at the monthly and annual time scale are considered for the

period of 2003-2015 with the focus on the world's major aquifers (Margat and der Gun, 2013). Monthly correlations represent how well seasonality is captured, while annual correlations measure the reproductions of secular trends as a result of interannual climate variability and groundwater depletion. It is important to note that TWS anomalies are not groundwater storage anomalies. However, interannual variation of TWS is heavily influenced by storage changes in the groundwater system (e.g. Scanlon et al., 2021).

## 3    Results and discussion

### 3.1  Pre-processing for transient simulation

#### 3.1.1 Node selection procedure

Figure 9a shows the performance for a HPCG strong scaling experiment (see Section 2.1) on a single thin node of the Dutch national supercomputer Snellius (see Table 3), up to a maximum of 128 cores using two CPUs. In this figure, the ideal performance (dashed line) is a straightforward extrapolation of the serial performance. It can be observed that a flattening occurs starting from 32 cores, where the competition of cores for the memory bandwidth results in saturation. From this, the maximum number of cores per node is chosen as $N_{CPN} = 32$ for the remainder of the experiments. For the node selection procedure (see Section 2.3.1, Figure 4b), the Americas model is chosen ($j = 2$) considering a one-year simulation for 1958. This model corresponds to a "medium-sized" model consisting of $N_{cell}^2 = 77$ million cells. Starting with $N_{nod}^2 = 1$ node, hence using a total of $N_p^2 = 1 \cdot 32 = 32$ processor cores, Figure 9b shows that for the third iterate the measured performance exceeds the target performance, $R^2 = 145 > 117 = R_{tgt}$, hence $N_{nod}^2 = 3$ nodes for a total of $N_p^2 = 3 \cdot 32 = 96$ cores. Therefore, the target submodel size is $B_{tgt} = N_{cell}^2 / N_p^2 = 77 / 96 = 0.8$ million $30''$ cells. Then, by computing $N_{nod}^i = \left\lceil N_{cell}^i / B_{tgt} \right\rceil$ for $i = 1, \ldots, 4$, and calculating back the number of nodes and cores, the Afro-Eurasia model is estimated to use seven nodes and 224 cores to meet the target performance, the Americas model three nodes and 96 cores, the Australia and Islands model each uses a single node and 32 cores, see Table 4. Hence, the total maximum number of nodes in the study for the GLOBGM is $N_{nod} = 12$ for a total of 384 cores. This results in an average submodel size of 0.72 million $30''$ cells.

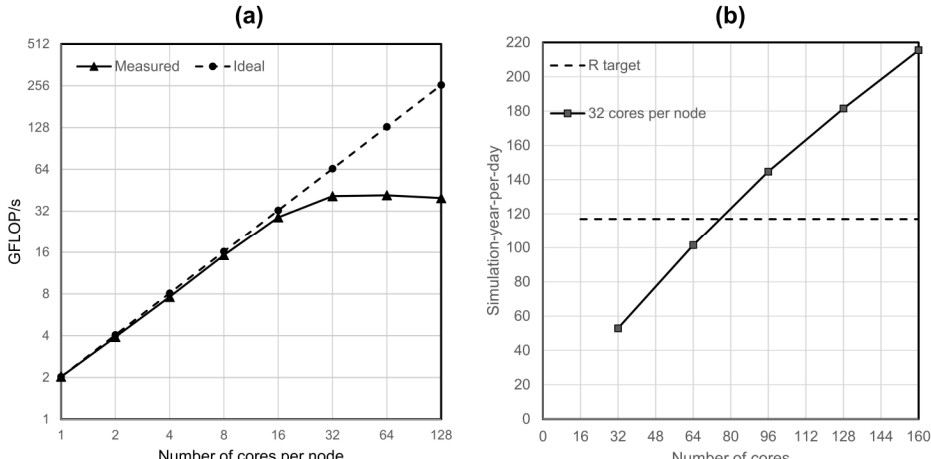

**Figure 9.** (a): HPCG performance results in GFLOPS/s for a model having $104^3$ cells and a maximum runtime of 60 s, (b): Performance estimation for the Americas model considering a transient simulation for 1958.

**Table 4.** Configuration of nodes and cores resulting from the node selection procedure.

| Model | $i$ | Number of Cells (M) | Submodel Size (M) | Number of Nodes | Number of Cores |
|---|---|---|---|---|---|
| Afro-Eurasia | 1 | 167.51 | 0.75 | 7 | 224 |
| Americas | 2 | 77.13 | 0.80 | 3 | 96 |
| Australia | 3 | 16.34 | 0.51 | 1 | 32 |
| Islands | 4 | 17.35 | 0.54 | 1 | 32 |
| | | 278.33 | 0.72 | 12 | 384 |

### 3.1.2 Tiled parameters and model input

The parameter pre-processing for the 163 data tiles (see Section 2.3.2) is distributed over the 12 available nodes, such that the first 11 nodes do the pre-processing for 14 tiles each, and the last node for 9 tiles. For each tile, the average runtime for pre-processing 1958-2015 (696 stress-periods) is 3 hours 25 minutes. In serial, this would require 558.7 core hours in total or ~23 days of runtime accordingly. This results in 5,578 PCRaster files ($10 + 696 \cdot 8 = 5,578$; see Table 1) for each tile, requiring 68 GB of storage. Hence, in total 909,214 files are written in parallel, amounting to 10.8 TB storage. Using data tiles therefore

saves storing ~8 TB of redundant data (43% reduction; see Section 2.3.2).

    Table 5 shows the pre-processing runtimes for each submodel to generate the transient MODFLOW input data in parallel using the node configuration as in Table 4, considering straightforward grid cell partitioning, see Section 2.1. On average, it takes up to half an hour to do the pre-processing for each Afro-Eurasia, Americas and Australia model. As can be observed, clearly not all submodels require the same runtime and there is a spread in distribution. Looking at the standard deviation (STD

in Table 5), this is significantly largest for the Islands model, measuring about 2.5 hours. We believe this is inherent to our chosen partitioning for the Islands model, allowing submodels to have grids cells of many scattered islands that are scattered across the world, see e.g. Figure 10 for the slowest submodel. By this, it is likely that random data-access of scattered unstructured grid cells to the tiled parameter grids occurs, slowing down the data reading after the submodel assembly. Although pre-processing is not a focus in this study, we might improve this in the future by incorporating a clustering constraint

in the partitioning strategy. For the Afro-Eurasia, Americas and Australia models, the STD is comparably small varying from 3 to 9 minutes. For the total parallel pre-processing to generate model input, a total of 222 core hours was required or ~9 days of serial runtime accordingly.

**Table 5.** Pre-processing runtimes for process "Assemble & Write Submodel Input Data", see "B" in Figure 7. STD = standard deviation.

| Model | Average | Min. | Max. | STD | Core hours |
|---|---|---|---|---|---|
| Afro-Eurasia | 00:27:13 | 00:20:44 | 00:46:50 | 00:05:04 | 101.6 |
| Americas | 00:28:42 | 00:22:41 | 01:40:47 | 00:09:06 | 45.9 |
| Australia | 00:20:08 | 00:16:54 | 00:27:10 | 00:03:20 | 10.7 |
| Islands | 01:59:52 | 00:26:48 | 15:00:46 | 02:25:40 | 63.9 |
| | | | | | 222.2 |

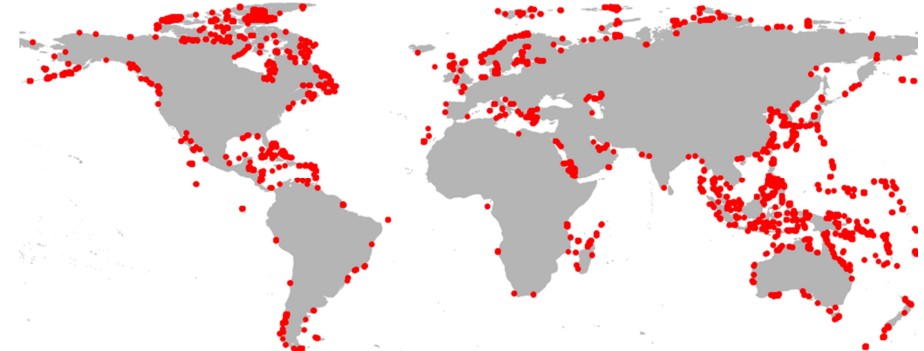

**Figure 10.** Location of unstructured grid cells (red) for a submodel of the Islands model, showing scattering, and requiring large pre-processing run-time for writing the submodel input data.

## 3.2 Parallel performance for transient simulation

Figure 11 shows the main parallel performance results for the GLOBGM, considering simulation of 1958-2015 including a 20-year spin-up using the node/core configuration as in Table 4, and applying straightforward grid cell partitioning (see Section 2.1.4). This figure shows that the target performance of 117 SYPD (16 hours of runtime) was achieved for all models (green bars), showing a significant increase in performance compared to the serial case (red bars). For the largest Afro-Eurasia model, the serial runtime could be reduced significantly from ~87 days (0.9 SYPD) to 0.63 days (or ~15 hours; 123.6 SYPD) using 224 cores. This corresponds to a speedup factor is 138.3 with a parallel efficiency of 62%. In general, for all four models, the parallel efficiency is ~60%. Note that Figure 11 also shows the metric core-hours-per-simulated-year (CHPSY) on the right vertical axis, that is defined as the cumulative runtime over all processor cores being used for simulation a single year. With CHPSY, the actual parallel runtime can be easily obtained by multiplying CHPSY with the number of simulated years and consecutively dividing by the number of processor cores being used.

It should be noted that the serial performance was only evaluated for a single year, i.e., 1958. The reason for this is that serial runtime for the entire simulation period was too long (e.g., ~27 CHPSY for Afro-Eurasia, hence $27 \cdot 78 = 2,106$ hours or 87 days runtime) and it would exceed the maximum allowed runtime of 5 days on a single Snellius node. For each parallel run, however, the performance is evaluated for 78 years (20 times 1958 + 1958-2015). Furthermore, in our experiments each serial/parallel run is only evaluated twice, taking average performance values for two runs only. We therefore did not account for any statistic (hardware related) runtime variation on the Snellius supercomputer. The reason for this was the limited total number of available core hours, where one full GLOBGM run requires ~24,000 core hours.

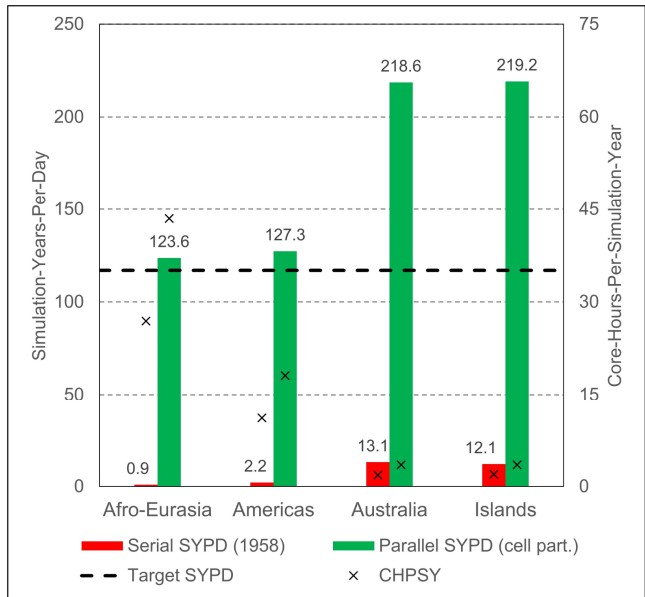

**Figure 11.** Parallel performance results for the models of the GLOBGM, considering straightforward grid cell partitioning ("cell part."). Left vertical axis: SYPD (bars); right vertical axis: core-hours-per-simulated-year CHPSY (crosses). Serial performance is computed for 1958 only.

Figure 11 shows that there is a performance difference between the models. The slowest model is the Afro-Eurasia model, followed by the Americas model, the Australia model and Islands model. This holds for both the serial and for the parallel case. For the serial performance it obvious that the main difference in performance is caused by the difference in model sizes. The serial performance difference between the largest Afro-Eurasia model having 168 M cells and the smallest Australia model having 16 M cells, that is 10 times smaller, can be directly related to lesser FLOPS and I/O.

However, also the difference in total number of iterations (linear + non-linear) contributes to this: considering 1958, the serial Afro-Eurasia model required 1.47 times more iterations to converge than the Australia model. The difference in iterations is likely related to the problem size and the number of the Dirichlet boundary conditions effecting the matrix stiffness. For the parallel case, this effect is enhanced, since the parallel linear solver has different convergence behavior and generally requires more iterations to converge when using more submodels, inherent to applying the additive Schwarz preconditioner. In general, for the parallel performance considering straightforward grid cell partitioning, the increase in the total number of iterations is in the range of 34% to 58%. That directly relates to a significant performance loss. Although we found that the parallel performance is adequate to reach our performance goal, reducing the total number of iterations is therefore something to consider for further research. We might improve the number of iterations by tuning the solver settings or applying a more sophisticated paralleled preconditioner. However, in general, users do not spend time on tuning solvers settings, and we therefore take them as they are. Furthermore, as a first attempt to reduce iterations, we did not see any improvement by using the additive coarse grid correction preconditioner.

Moreover, the difference in parallel performance could be explained by the differences in submodel (block) sizes (see Table 4). Due to rounding the number of cores to the number of cores per node, the block sizes for the Australia and Islands models are ~1.5 smaller than the block sizes for the Afro-Eurasia and Americas models, which directly results in an increasing performance.

Besides the increase of in iterations, memory contention contributes to the loss of parallel performance (see Section 2.1.5), even when using 32 cores per node out of 128 cores. From the HPCG test, see Figure 9a, the parallel efficiency using 32 cores per node is 63%, which is likely to be a representative value for the MODFLOW linear solver. For MODFLOW, however, this value is likely to be slightly larger because of non-memory bandwidth dependent components. Comparing to a 1958 run for the Americas model, and using one core per node, we estimate the maximum efficiency to be 77%. Extrapolating this to

the Afro-Eurasia model, this means that ~172 cores could be used efficiently, increasing the efficiency to 80%. This value is more in range with we expect from preceding research using MODFLOW (Verkaik et al., 2021b).

In Figure 12a, the performance results are given for catchment partitioning considering HydroBASINS Pfafstetter levels 8, 6 and 5 (see Section 2.1) chosen for illustration. Level 7 was excluded deliberately in the search for finding a significant performance decrease and minimizing the allocated budgets on the Snellius supercomputer. In general, for the Australia and Islands model, the target performance is exceeded for all HydroBASINS levels. For the Afro-Eurasia and Americas models,

level 8 is sufficient to reach the target and level 6 results in performance slightly below the target. For these models, level 5 results in about three quarters of the target performance. In general, using HydroBASINS catchments up to Pfafstetter level 6 seems to give adequate results. Except for the Australia model, performance decreases when the Pfafstetter level decreases. The reason why this doesn't apply to the Australia model is because of the slight iteration increase for level 8. In Figure 12b, the performance for catchment partitioning (Figure 12a) is normalized with the performance using straightforward grid cell

partitioning (Figure 11) and the associated total iteration count. Clearly, the performance slope is correlated to the inverse of the load imbalance determined by METIS, see Section 2.3.4. Since many factors may contribute to load imbalance, like the specific multilevel heuristics being used, specified solver settings, characteristics of the graph subject to partitioning, limitations of the specific software library, an in-depth analysis is beyond the scope of this study. Here, we simply assume that this imbalance is a direct result of the coarsening associated to decreasing Pfafstetter levels for a fixed number of submodels,

reducing the METIS graph-partitioning search space.

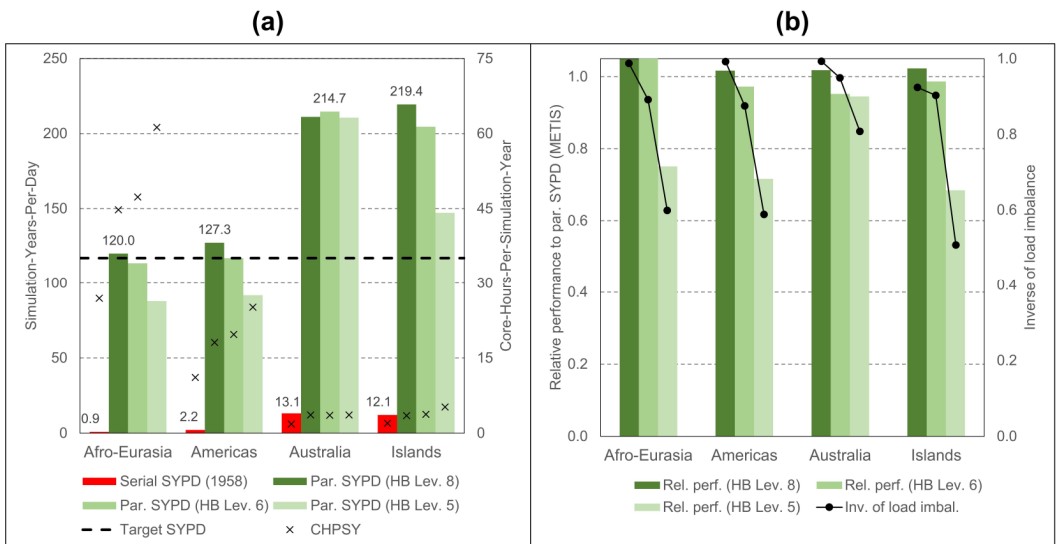

**Figure 12.** (a): Performance results for the GLOBGM, considering catchment partitioning using HydroBASINS (HB) catchment areas for
Pfafstetter level 8, 6 and 5. Left vertical axis: SYPD (bars); right vertical axis: CHPSY (crosses). Serial performance is computed for 1958 only. (b): Relative performance to performance using straightforward grid cell partitioning, normalized with the corresponding total number of iterations (bars), and inverse of the METIS load imbalance $I_i$ (lines).

The required storage for each run is given by Table 6, where monthly computed hydraulic heads were saved exclusively during simulation. For this, one transient global run required 8.8 TB of input and 1.4 TB of output.

**Table 6.** Input and output of the GLOBGM for simulating 1958-2015.

| Model | Input (TB) | Output (TB) |
|---|---|---|
| Afro-Eurasia | 5.27 | 0.85 |
| Americas | 2.53 | 0.39 |
| Australia | 0.44 | 0.08 |
| Islands | 0.56 | 0.09 |
| | 8.80 | 1.41 |

## 3.3 A limited first evaluation of the GLOBGM

### 3.3.1 Steady-state hydraulic heads for the CONUS

Figure 13 shows the steady-state water table depth residuals aggregated to the HUC4 surface watershed boundaries for the CONUS, comparing the GLOBGM to the GGM (de Graaf et al., 2017), the GIM (Fan et al., 2017) and the CGM (Zell and Sanford, 2020), see Section 2.4.1. For this Figure, all NWIS measurements from Zell and Sanford (2020) are considered, for both sedimentary basins and mountain ranges. In addition to Figure 13b, in Figure A1 in Appendix A the results for the GLOBGM and GGM are shown for sedimentary basins with and without karst, showing best results for sedimentary basins excluding karst as to be expected from the presumed application range (see Section 2.2.1).

Clearly, the GLOBGM is an improvement compared to the GGM, where the frequency distribution of the GGM residuals (red line) shifts towards the residuals of the GLOBGM (green line). Hence, refining from 5′ to 30″ resolution is resulting in higher and more accurate hydraulic heads. A likely explanation for this better performance is that the GLOBGM, having a higher resolution, is better in following topography and relief, in particular to resolve smaller higher altitude groundwater bodies in mountain valleys. Also, higher resolution models have a smaller scale gap with the in-situ head observations in wells. However, the GIM and CGM seem to give better performance than the GLOBGM. A reason for this could be that those models are calibrated, using many data from the Unites States, while the GLOBGM (and the GGM) is not. In general, the computed hydraulic heads with the GLOBGM still seem rather low. Analyzing the possible causes for this is left for further research.

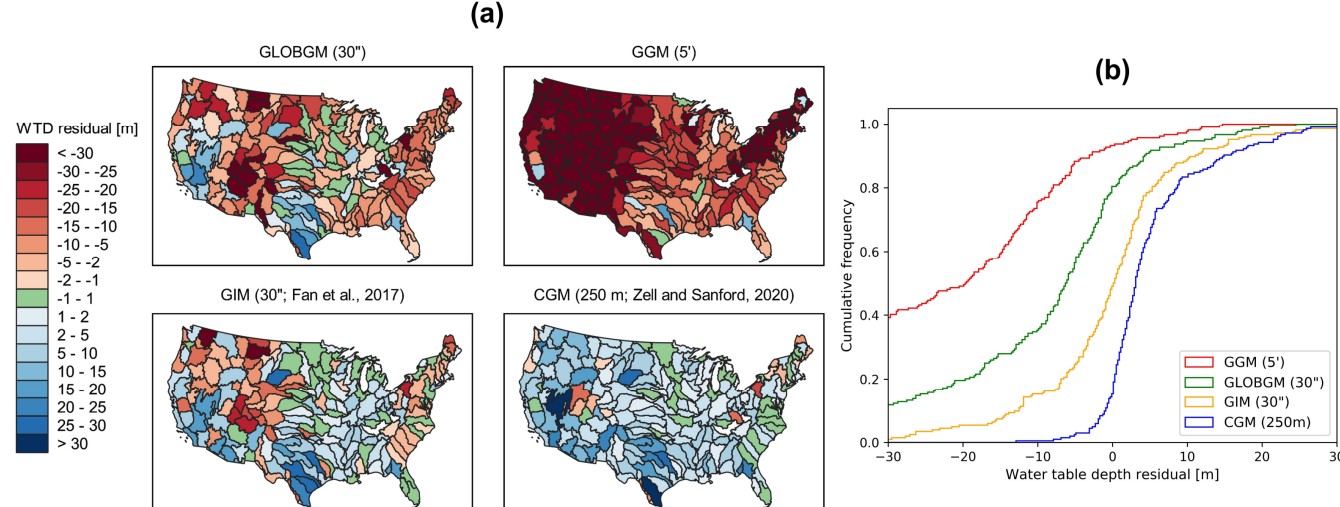

**Figure 13.** (a): Water table depth (WTD) residuals aggregated to HUC4 units for the steady-state GLOBGM compared to the GGM, the GIM model of Fan et al. (2017) and the CGM of Zell and Sanford (2020). (b): Corresponding cumulative frequencies.

### 3.3.2 Transient hydraulic heads for the CONUS

Figure 14 shows the results for the transient evaluation of the GLOBGM, comparing the computed transient hydraulic heads for 1958-2015 to the transient NWIS head observations. Three statistical measures are evaluated: average timing ($r_{mo}$), average amplitude error ($IQRE_{mo}$), and trend classification (using $\beta_y$), see Section 2.4.2. Furthermore, similar statistics are computed and added to this figure for the coarser 5′ GGM, showing the effect of increased resolution with the GLOBGM. In general, we see that the GLOBGM and the GGM give very comparable results that could be further improved. For the average amplitude error, the GLOBGM seems to perform slightly worse, for which we do not have a straightforward explanation. However, the GLOBGM seems slightly better regarding trend direction and lacking model trend. Furthermore, it can been that about 40-50% of the (majority of) obervations have a mismatch in trend. This can likely be related to incorrect well locations and pumping rates in the model, inherent to the applied parametrization and concepts within PCR-GLOBWB (see Table 1). This might also have effect on the discrepancies for the timing and amplitude errors and is therefore subject for further research.

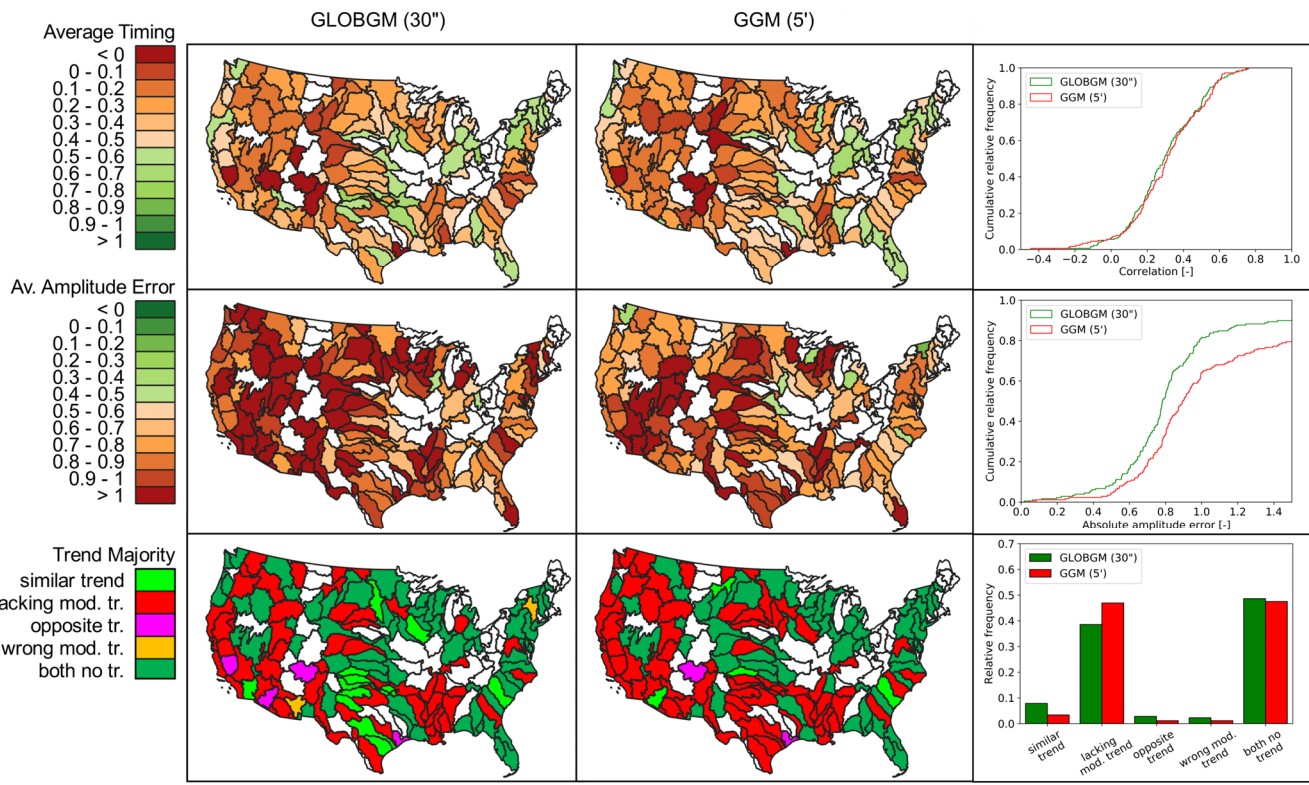

**Figure 14.** Evaluation of computed hydraulic heads for 1958-2015 to NWIS head observations: average timing (first row panels), amplitude error (second row panels) and trend (third row panels), for both the GLOBGM as well as the GGM. Plotted white colors for HUC4 units mean that less than five samples were found and hence was not considered representative.

### 3.3.3 Total water storage anomalies for the world's major aquifers

In Figure 15 we present the cross-correlations between GRACE total water storage (TWS) anomalies at monthly and annual time scale with simulated TWS anomalies calculated from the sum of the GLOBGM (groundwater component) and PCR-GLOBWB model (Sutanudjaja et al., 2018) (other components). These results demonstrate a good overall agreement between our simulated TWS and GRACE, especially at the monthly resolution (Figure 15a). At the annual resolution (Figure 15b), the agreement remains satisfactory, particularly for major aquifers known for having groundwater depletion issues, such as the Central Valley, High Plains Aquifer, Middle East, Nubian Aquifer System, and North China Plain. However, somewhat lower annual correlations are observed for the Indus and Ganges Brahmaputra, which could be attributed to factors such as glacier storage changes that the PCR-GLOBWB model does not accurately simulate. Furthermore, the low annual correlations observed in the Amazon may be attributed to the large water storages in floodplains during, which is not properly simulated in the used version of the PCR-GLOBWB model. Also, disagreement in South American and Africa can be attributed to the

PCR-GLOBWB forcing data issues in such regions with limited availability of meteorological observations. Additionally, it is important to consider that GRACE measurements may display a higher noise-to-signal ratio in arid regions like the Sahara, especially when there are minimal storage changes or low groundwater utilization, which could contribute to relatively lower correlations in those areas.

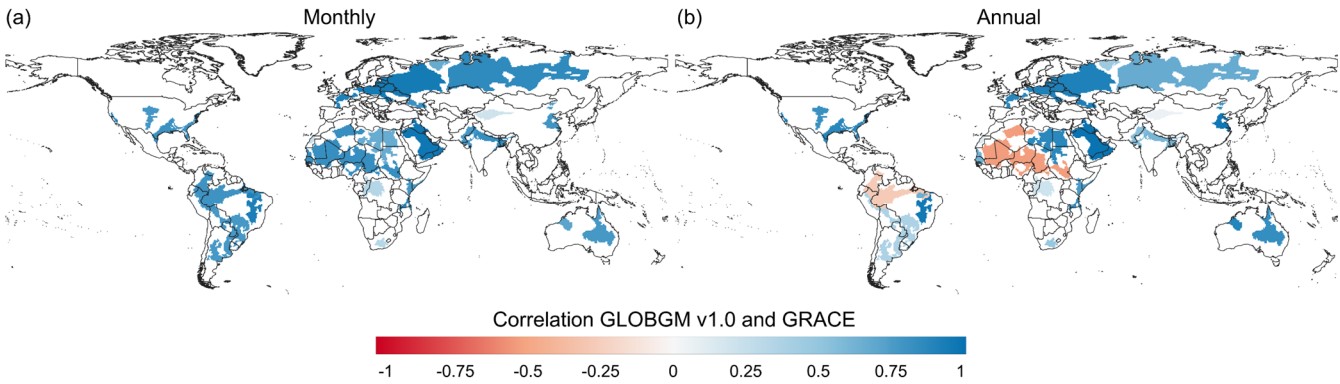

**Figure 15.** Correlation between monthly (a) and annual (b) TWS time series from GRACE and the ones computed from the GLOBGM (groundwater) and the PCR-GLOBWB model (snow, interception, soil moisture and surface water). Here we focus on the major aquifers.

### 3.4 Example of global-scale results

To provide an impression about the level of detail reached when simulating hydraulic heads at 30″ spatial resolution, steady-state and transient global map results are shown in Figure 16. For sake of the application domain of the GLOBGM (see Section 2.2.1) we mask out the karst areas (WHYMAP WoKAM; Chen et al., 2017). In Figure 16a, the steady-state solution of water table depths is shown: the panels with blowups show the intricate details present, which are mostly guided by surface elevation and the presence of rivers. In Figure A2 in Appendix A, cell locations are plotted where the steady-state GLOBGM has a vertical flow in upward direction to the land surface (groundwater seepage), mainly showing clustering near river locations, but also areas where heads underlying the confining layers exhibit an overpressure compared to the overlying phreatic groundwater. Figure 16b and Figure 16c show examples of transient global map results for the GRACE period 2003-2015, here focusing on sedimentary basins only. Figure 16b shows the hydraulic head amplitude as represented by the interquartile range, where the amplitude size reflects the amplitude of groundwater recharge and the hydrogeological parameters (storage coefficient and hydraulic conductivity). Figure 16c shows the trends of yearly average hydraulic heads for sedimentary basins. For the areas with confining layers, the trend in the heads of the lower model cells is taken. Here, the well-known areas of groundwater depletion (Wada et al., 2010; de Graaf et al., 2017) are apparent. However, also areas with spurious trends can be seen that may be connected to incomplete model spin-up. Furthermore, positive trends can be seen, which may be connected to increased precipitation related to climate change. In the supplement to this paper, we provide two animations for computed water table depths: one for monthly values and another one for yearly averaged values. To show the dynamic behavior of the hydraulic heads, we show these results relative to the heads of 1958.

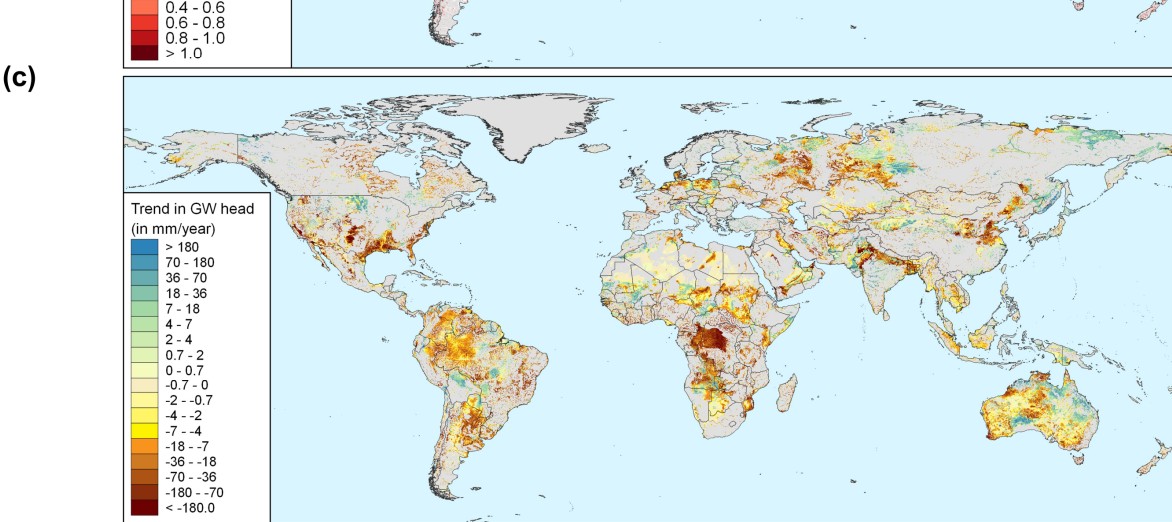

**Figure 16.** Example outputs of the GLOBGM. (a): Steady-state water table depth with detailed blow ups; (b): Water table amplitude for 2003-2015 as represented by the interquartile range IQR. (c): Hydraulic head trends for 2003-2015 for the aquifer (lower model layer). In all panels a mask is applied for karst areas; in the lower panels (b) and (c) an additional mask is applied for mountain ranges.

# 4 Conclusions and recommendations

The PCR-GLOBWB-MODFLOW global-scale groundwater model at 30″ spatial resolution (GLOBGM v1.0) was successfully implemented using high performance computing for simulating long transient periods. By this, we demonstrated that refining the PCR-GLOBWB-MODFLOW model from 5′ resolution to 30″ resolution is computationally possible. This can be seen as a small but important step towards better global-scale groundwater simulation. To our knowledge, the GLOBGM is the first implementation of a transient global-scale groundwater model at 30″ resolution. Our implementation uses unstructured grids to cancel redundant sea and land cells, and effectively applies a parallelization approach that organizes the global model as a set of independent parallel models, resulting in three continental-scale groundwater models (Afro-Eurasia, Americas, and Australia) and one remainder model for all (smaller) islands. We showed that our workflow, using parallel pre-processing and a new parallel distributed memory prototype version of MODFLOW 6, is effective for achieving a user-defined parallel runtime target and to minimize data usage. This is demonstrated for an experiment on the Dutch national supercomputer Snellius, simulating the GLOBGM for 1958-2015, considering both grid cell partitioning and catchment partitioning.

With our approach, we first estimated the required node/core configuration on the Snellius supercomputer to achieve a set target of 16 hours runtime including the 20-year spin-up (leading to 117 Simulation-Years-Per-Day; SYPD), and then we conducted the parallel pre-processing and illustrated its necessity. For grid cell partitioning, we showed that a maximum of 12 nodes running with 32 cores per node is required to meet the target for each of the four underlying groundwater models of the GLOBGM. For the largest Afro-Eurasia model using seven nodes (224 cores), runtime was reduced from ~87 days to 15 hours (124 SYPD). For catchment partitioning, the first results presented here are promising. Using HydroBASINS catchments as an example, we showed that this lesser optimal partitioning results in quite similar parallel performance down to Pfafstetter level 6. Since our implementation is suitable for parallel systems with relatively limited hardware requirements, we believe it is well suitede for users who do not have exclusive access to many nodes and need to deal with queuing times. We therefore believe that our implementation will contribute to future model improvements.

Although the main purpose of the paper was to show that 30″ global transient groundwater simulations are possible at reasonable computational costs, we also performed a limited model evaluation. From comparison with NWIS head observations for the CONUS, we conclude that the steady-state hydraulic heads from the GLOBGM are significantly better compared than those from the 5′ PCR-GLOBWB-MODFLOW model, but still could be improved compared to measurements and model results from Fan et al., (2017) and Zell and Sanford (2020). For the transient simulation, results for the GLOBGM and the 5′ model are comparable for the CONUS, both giving significant differences compared to measurements. Monthly and multi-year total terrestrial water storage anomalies for major aquifers around the world, as derived from the GLOBGM and PCR-GLOBWB model, compared favorably with observations from the GRACE and GRACE-FO satellites. Although the exact reasons for the differences with head measurements are kept for further research, along with further model improvement, they are likely a result of spatial difference in resolution, lacking transient model input data, e.g. for groundwater well abstraction, or must be found in improving the hydrogeological schematization.

Although the current parallel performance is quite satisfactory for its purpose, it could be further improved, e.g. by improving the processor core utilization, improving the parallel preconditioner for the linear solver to account for the increasing number of iterations, and reducing the pre-processing times. First, multicore CPUs are likely to have more and more memory channels to be more applicable to memory-bound problems and we expect better core utilization with next generation processors. Second, model iterations could be reduced likely by tuning the MODFLOW solver settings or improving the parallel preconditioners. Third, parallel pre-processing runtimes could possibly be reduced by improving the partitioning, e.g. by clustering many smaller islands causing random access data patterns for the Islands model.

Regarding storage, users should be aware that GLOBGM requires more than 21 TB (900k files) of data for a single run. Since we now exclusively use uncompressed PCR-Raster files, requiring a large amount of storage, compression could be considered for follow-up research, as well as using more data tiles.

Applying catchment partitioning gives opportunities for further research, e.g. to realize a parallel coupling to the PCR-GLOBWB surface water routing module with acceptable parallel performance for the groundwater model, or to apply such grid-independent partitioning to even higher resolutions.

*Code and data availability.* The GLOBGM v1.0 is open source and distributed under the terms of GNU General Public License
v3.0, or any later version, as published by the Free Software Foundation. The model tools and data are provided through the GitHub repository https://github.com/UU-Hydro/GLOBGM.git (Verkaik and Sutanudjaja, 2022, https://doi.org/10.5281/zenodo.7398200), where the main model data can be accessed through https://doi.org/10.24416/UU01-44L775. The parallel kernel of MODFLOW 6 used for the GLOBGM v1.0 is provided through https://github.com/verkaik/modflow6-parallel.git (Verkaik et al., 2021c, https://doi.org/10.5281/zenodo.5778658). This kernel
is open source through the CC0 1.0 Universal public domain dedication. Although development and maintenance of the official version of the GLOBGM is conducted at the Department of Physical Geography, Utrecht University, we welcome and encourage researchers from external parties to contribute.

*Author contributions.* JV performed the conceptualization, methodology, and implementation for all workflows, as well as the
810 parallelization of MODFLOW 6. EHS helped with the conceptualization of this research and prepared all of the GLOBGM raster data for the transient experiment. JV performed all other pre-and post-processing, simulation on the Dutch national supercomputer Snellius, and analysis of all results. MFPB, HXL and GHPOE supervised this research and helped with its conceptualization. JV prepared the manuscript with contributions from all authors.

*Competing interests.* The authors declare that they have no conflict of interest.

*Acknowledgements.* We thank Joseph D. Hughes and Christian D. Langevin for their valuable suggestions and comments on parallelizing MODFLOW. We also thank Deltares for making this research possible. This work was carried out on the Dutch national e-infrastructure with the support of SURF Cooperative. MB acknowledges support from the European Research
Council, ERC AdG grant GEOWAT under grant agreement No: 101019185.

## Appendix A

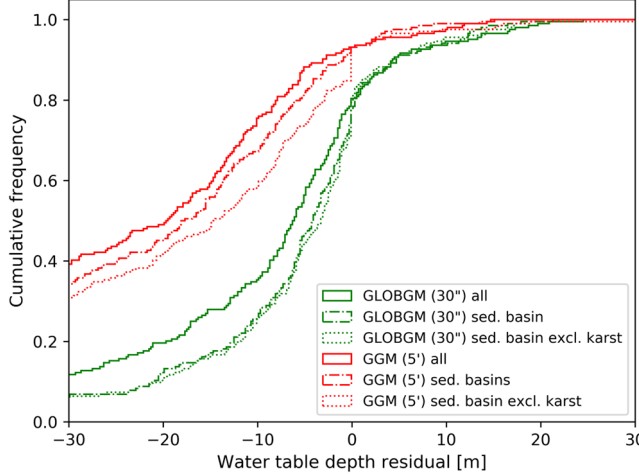

**Figure A1.** Cumulative frequencies of water table depth residuals aggregated to HUC4 units for the steady-state evaluation considering the GLOBGM (green lines) and the GGM (red lines). Solid line: including all NWIS wells, similar as in Figure 13b; dash-dotted line: for NWIS wells only in sedimentary basins; dotted line: for NWIS wells only in sedimentary basins excluding karst.

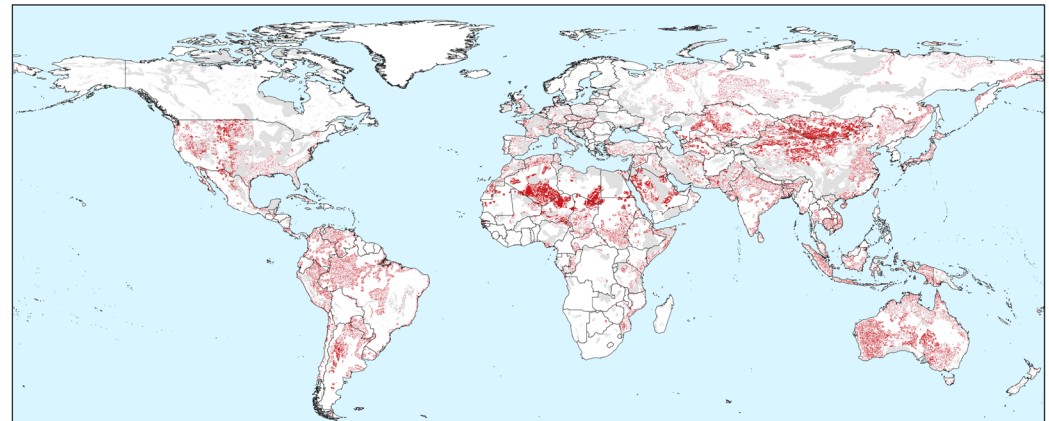

**Figure A2.** Locations (red) of where groundwater seepage occurs from the lower model layer towards the upper model layer.

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
