# Peer review of "GLOBGM v1.0: a parallel implementation of a 30 arcsec PCR-GLOBWB-MODFLOW global-scale groundwater model"

_Geoscientific Model Development, 2022_

## Author Comment (AC3)

To: Editors of Geoscientific Model Development

**Faculty of Geosciences**
Department of Physical Geography
PO Box 80.115, 3508 TC Utrecht, The Netherlands

**Visiting address**
Vening Meinesz A, Princetonlaan 8a
3584 CB, Utrecht, The Netherlands

**Website**
www.uu.nl/geo/fg
**Email** jarno.verkaik@deltares.nl
**Phone** +31(0)6 4691 4636

**Date: April 18, 2023**
**Subject: Final Response**

Dear Editor, Dear Wolfgang Kurtz,

We would like to thank you for your time for reading our manuscript "GLOBGM v1.0: a parallel implementation of a 30 arcsec PCR-GLOBWB-MODFLOW global-scale groundwater model". We would also like to thank the two anonymous reviewers for their time for thoroughly reading our manuscript and coming up with valuable comments as posted during the open discussion. We were pleased to read their positive feedbacks on our paper's main goal, which is to tackle the technical challenges for parallelizing our global groundwater model, as we state in line 67 of our manuscript: "*In this paper, the focus in on the technical challenges of implementing the GLOBGM using HPC*". We believe that from this perspective, the findings presented in our paper would appeal your readers, supported by the reviewers' comments "*The paper is well written and interesting from a computational perspective. I do not see any problems with the technical steps.*" (Reviewer 1), "*The manuscript is well-organized and successfully demonstrates a computational workflow... For this reason, it is an important contribution.*" (Reviewer 2). Besides reaching readers that are specifically interested in global groundwater modeling, we also believe that our presented methodology is interesting for readers that have large regional-scale or local-scale MODFLOW 6 groundwater models that need significantly faster computations, since our methodology is in essence not restricted to global modeling exclusively.

Although we are pleased to read these positive feedbacks, we also recognize that Reviewer 1 is critical about the usefulness and application range of the presented global model and of global models in general. Although we believe that we have been explicit about the main goal of this paper (parallel implementation of a global groundwater model needed for increased resolution) and also that many model improvements are still needed for our initial version 1.0 of our GLOBGM (e.g., in line 36 "*However, results for the transient simulation are quite similar and there is much room for improvement.*"), we welcome these comments and the valuable suggestions made by both reviewers and will take them into account for a future version. To accommodate both reviewers in their comments about the *global model performance* in this paper, we suggest updating the manuscript as follows (see in more detail our replies to the reviewers):

1. *Introduction:* Based on literature, we will add a paragraph to the beginning of Section 1 about which developments are needed to improve global groundwater models beyond the current state-of-the-art. This will also identify as one these improvements increasing the resolution and increasing the speed of

[Figure]

computation to make this possible. This will then immediately make clear which of these possible improvements are the focus of this paper.

2. *Methods:* In Section 2.2.1, "Description of the GLOBGM", we will add a paragraph about the assumptions and limitations of our GLOBGM model with respect the work for the 5′ global groundwater model, De Graaf et al. (2015, 2017). In this, we will again be more explicit that our work addresses *the computational challenges need to be overcome if the resolution of this model is increased.*

3. *Steady-state model evaluation:*
   3.1. We will add extra curves for our GLOBGM to Figure 12b showing the errors for sedimentary basins without karst, sedimentary basins and karst and non-sedimentary basins.
   3.2. We will mask out the karst areas in the steady-state global maps in Figure 14a.
   3.3. We will change the legend of in Figure 14a to allow for groundwater levels and heads above the surface.
   3.4. We will add a separate figure to Figure 14 showing the heads in the confined aquifers, and mask out the Karst areas.

4. *Transient model evaluation:*
   4.1. We will mask out the karst areas for the global maps of Figure 14b and Figure 14c.
   4.2. We will change the legend of Figure 14c for showing both positive and negative trends.
   4.3. We will add a supplemental document explaining the filtering of transient well locations in more detail.
   4.4. We will investigate the possibilities for comparing with GRACE data for this version of our GLOBGM, version 1.0, or an upcoming version.

Based on the comments of Reviewer 2, we suggest making the following additional changes in the manuscript regarding the technical implementation of the GLOBGM parallelization and typographical/syntactical errors (see in more detail our reply to Reviewer 2):

5. We will add text to Section 2.1 for better explaining the purpose of sorting for the global domain decomposition.
6. Furthermore, in this Section, we will add text to improve the connection of the example of Figure 2 to the actual global domain decomposition being used.
7. In Section 2.3.1, we will add text to emphasize better the relationship between the "Model Workflow" and the "Node Selection Workflow".
8. We will include the typographical/syntactical suggestions and corrections.

We hope that our suggested changes in our manuscript are adequate for you to consider our manuscript for publication. Please let me know if you have any questions or more clarification if required.

On behalf of all authors,

With kind regards,

Jarno Verkaik

[Figure]

**Reply to comment of Reviewer 1 om GMD-2022-226**

**General reply**

We thank the reviewer for taking the time to evaluate our manuscript and the extensive comments. The reviewer states that the paper is well-written and does not see any problems with the technical steps of the of the paper. It is important to note this first, because the explicit goal of this paper is (line 66): *"the focus is on the technical challenges of implementing the GLOBGM using HPC"*. To our opinion, the merits of this paper should be evaluated based on that focus.

However, reviewer 1's objections against this paper are predominantly entirely based on the fact that the reviewer does not believe in the usefulness of global (we presume groundwater) models, given their current state of development and that building these and reporting on their progress should not be done. This is where we greatly disagree with the reviewer. There are two main reasons to disagree about this:

1. In any field of earth system science, modelling efforts started with grossly inaccurate first versions of models. We give three examples. the first global circulation models and numerical weather prediction models were very simple in their dynamics, model physics and parameterization (see e.g. Manabe, 1969). The first global vegetation models were simple and not at all predictive at the local scale (Prentice et al., 1992). The first global hydrological models (Alcamo et al., 1997) used very simple conceptual hydrology at low resolution, not at all capable of predicting streamflow at a specific catchment. Here is the question: If global modelling of a certain part of the earth system is deemed useful, should we wait until we are far enough to simulate all states of this system credibly everywhere, or should we start with what we know and develop from there? We firmly believe that in order to make progress, it is better to do the latter as otherwise chances are that there will be no progress at all. All progress must be done in small steps, just as climbing to the next floor needs a stair with 20 steps instead of two.  In fact, if imperfect global models cannot be built and developed upon, as suggested by reviewer 1, many of the papers in GMD would not have been published.
2. Even if global models are far from perfect in quite some aspects or regions, they can still be useful. The first global climate models, especially when multiple of these were used, were all good enough to provide order of magnitude estimates of the regional impacts of $CO_2$ increase. The first numerical weather prediction models extended the forecast horizon several days beyond simply nowcasting pressure fields by hand. The first vegetation models were very helpful in translating local time-series of pollen-based observations to spatiotemporal maps of vegetation shifts from the last glacial maximum into the Holocene. The first hydrological models served very well in pinpointing regions of current or emerging water scarcity.

This is not different when talking about global groundwater models. There is an obvious need by the earth system science community for building these, as testified in Condon et al. (2021). And this has led to the first so-called gradient-based groundwater representations for global hydrology (Fan et al., 2013; de Graaf et al., 2015; Reinecke et al., 2019). These mentioned authors developed the models and extensively tested their accuracy with head observations, showing their inaccuracy, particularly when estimating groundwater depth. And as inaccurate as they can be, they have already proven their worth in understanding the order of magnitude of, e.g., the importance of shallow groundwater in sustaining dry season evaporation (Miguez-Macho et al., 2012), the control of groundwater on rooting depths (Fan et al., 2017) and the impact of groundwater pumping on groundwater depletion (De Graaf et al., 2017) and the violation of environmental flow limits (De Graaf et al., 2019).

[Figure]

When it comes to the steps needed to make a leap change in improving the current generation of models, many steps are still needed; see also the reviews by Condon et al. (2021) and Gleeson et al. (2021). Specifically these are: 1) improved hydrogeological schematization, particularly including multilayer semi-confined aquifer systems and the macroscale hydraulic properties of Karst and Fractured systems; 2) increased resolution to better resolve topography and in particular resolve smaller higher altitude groundwater bodies in mountain valleys; 3) improved knowledge on location, depth and rate over time of groundwater extractions; 4) better estimated groundwater recharge, especially in drylands and at mountain margins; 5) increased computational capabilities to be able to make simulations with the above improvements possible. Our paper specifically revolves around items 2 and 5: *If we improve spatial resolution, how should we make this computationally possible?* It is a small but important step to better global groundwater that needs to be taken to proceed further. We recognize that the above-mentioned argumentation has not been explicitly mentioned in the manuscript so far; for that we thank the reviewer 1 for putting emphasize on his concern. Therefore, in the manuscript, we will provide a statement to this angle in the introduction of the paper to show how this paper fits in the many steps needed.

The paper presented here uses the global hydrogeological schematization of De Graaf et al. (2017) and inspects *what computational challenges need to be overcome if the resolution of this model is increased*. Thus, since the objections against this paper by reviewer 1 are really about the global model itself, not about the numerical techniques displayed here, the reviewer objections are in fact about the De Graaf et al. (2017) paper. In that De Graaf et al. paper, it has been explicitly stated that 1) they simulate the top aquifer systems, so unconfined aquifers and the uppermost confined aquifers; 2) although global results are shown, the model is mostly capable for the sedimentary alluvial basins (main productive aquifers). Since we take the De Graaf et al. (2017) schematization as input, these limitations also count for the model reported here. We will make this explicit clear in the revisited manuscript.

Where we do agree with reviewer 1 is that we should be open about the assumptions and limitations of the model. We presumed, since the focus was on the numerical scheme, that we did not need to extensively discuss limitations already discussed in the De Graaf et al. (2015, 2017) papers. However, as we agree with reviewer 1 that this is important, we will add a paragraph to section 2.2.1 explaining this more in detail. Particularly we will state that: The application domain of the model is as follows: 1) it is intended to simulate hydraulic heads in the top aquifer systems, so unconfined aquifers and the uppermost confined aquifers; 2) wherever there are multiple stacked aquifer systems, these are simplified in the model to one confining layer and one aquifer; 3) the model schematization is suitable for heads in large sedimentary alluvial basins (main productive aquifers) that have been mapped at a 5-arcminute resolution; 4) in as far these sedimentary basins include karst, it is questionable of a Darcy approach can be used to simulate large-scale head distributions; 5) due to the limited resolution of the hydrogeological schematizations, in mountain areas we simulate the heads in the mountain blocks but not those of groundwater bodies in hillslopes and smaller alluvial mountain valleys; 6) also, for the heads in the mountain blocks, we assume that secondary permeability of fractured hard rock can also simulated with Darcy groundwater flow; an assumption that may be questioned.

Replies to specific remarks

In the following we cite specific remarks by the reviewer in italics and provide our remarks in roman.

[Figure]

*For example, the steady state comparison with Fan, CMG GGM in Figure 12: The comparison shows that all these results are inconsistent. The authors mention an improvement to GGM, but all of these products are highly uncertain themselves.*

We do not agree that all results are inconsistent. It seems that we have not been clear enough about the model comparison, so we will improve that description in the revised paper version. We do compare between models, but we compare the differences between each model and the mean water levels in 34k well locations across the US. The models that perform best, i.e. the Fan et al. (2013) model and the Zell and Sanford (2020) USGS model, have been calibrated, while the 5-arcminute model of De Graaf et al (2019) and our 30-arcsecond model have not. This is a fair comparison of model capabilities. It also shows that especially the Zell and Sanford (2020) USGS model is the only one performing well in the mountain hard rock regions because of a) 250 m resolution that resolves many more groundwater pockets in mountain valleys; b) local calibration of transmissivities.

*But the section 426-430 is very hard to read. It is a very long list how the authors worked around data gaps, with no explanation why they do so, and there is no assessment to what extent these chosen steps are reliable. To be scientifically sound, every step has to be explained and demonstrated how, where and when the assumptions hold up. It is crucial and a basic scientific principle to root any model in reality.*

We regret that this has not been clear. We will add a supplemental where we will present more in detail how the selection of the locations with wells with groundwater level time series was done from the original NWIS dataset. We will also add the number of wells that remained after each filtering step. This was not a way to work around data gaps, but to retain only locations with sufficiently long time series and to ascertain that the filter of each groundwater observation well was assigned to the right model layer. We like to stress that even though actual model building is not the target of this paper - this is the introduction of an efficient scheme that makes high-resolution global groundwater modelling a reality- we have added a thorough evaluation of simulated depth to groundwater. Even though this was not global, given the huge variation of landscapes and hydrogeological settings in the U.S., we feel that this can be representative for also other world regions.

*For continents where no transient head data are available a comparison with changes of water table obtained through GRACE could be informative in the context of Figure 14. Such a comparison will clearly show that the model cannot reproduce the decline of the water table on a global scale.*

Thanks for this suggestion. GRACE would be an interesting means of comparing the results of the global groundwater model with. However, GRACE provides total terrestrial water storage anomalies, which includes many terms besides groundwater. So, to apply GRACE we need to correct for non-groundwater-related storage changes. We might do that with the results of the global hydrological model which is used to force our groundwater model: PCR-GLOBWB. But even then, there will be both positive as well as negative trends based on a combination of groundwater withdrawal and climate variability. We will investigate the possibilities for comparing with GRACE data for this version of the GLOBGM version 1.0 or an upcoming version.

*The authors acknowledge many areas of improvement but, to highlight this point again, fall short in clearly declaring that none of the results should be used in any other context than in a software development framework.*

[Figure]

As we argued above, we feel that just because global models are in earlier stages of development does not mean that they cannot be used for certain analyses. So, we will not make this statement. As to illustrate a possible application of this global groundwater modelling: when comparing the model runs with and without groundwater withdrawals over the last 60 years, the *large-scale* impacts of groundwater use on the global heads, groundwater storage and streamflow can be assessed, while taking into account groundwater-surface water interaction, increased capture and later groundwater flow (as cannot be done with water-balance based methods). This application of a global groundwater was shown before in De Graaf et al. (2017) and De Graaf et al (2019).

*The only reliable data source is topography, but through the rough discretization a lot of information is lost, especially in steep terrains. The global geological products are speculative at best, a significant source of uncertainty. There are countless conceptual problems. Groundwater abstraction or rivers cannot be reliably simulated with these spatial resolutions in the MODFLOW conceptualization. The wells and rivers are cell-centered, making a robust simulation of drawdown cones or mounds extremely uncertain. The associated temporal dynamics of the water table decline cannot be captured with these resolutions. As the authors are presenting a transient model this is a fundamental problem.*

Yes, by using a 1 km discretization lot of information is lost that would be needed to make *local inferences*. But this is not the case for *sub-regional to regional assessments* at the global extent as is the primary goal of a global extent model. We disagree that groundwater abstractions and the impacts of rivers cannot be reliably simulated. This can be done if answers are needed at the subregional to regional scale and provided that hydraulic resistances between surface waters and the groundwater systems are scale-consistent. This has been done for over four decades with regional groundwater models all over the world. These regional models had lower resolutions than ours until well into the 1990s. Does this mean, that all these models were useless? Evidently, they were not as they were used extensively as a basis for water management and policy. Of course, that is not our goal, since we do not have subregional scale hydrogeological data. But we could if these were available. For instance, on a regional scale, we want to mention the Mekong Delta model (Minderhoud et al, 2017), which reproduces well enough land subsidence caused by groundwater extractions using a grid cell size of 1x1 km$^2$; the same resolution as our GLOBGM version 1.0. This model has been well-received by the local community and has led to changes in groundwater extraction policies under the so-called Decree 167, which prescribes restrictions on groundwater extraction in aquifers within the territory of the Socialist Republic of Vietnam. Regarding transient simulations: these can be done if the goal is to study water table or hydraulic head variations at scales larger than the model resolution. Again, this has been done for decades with regional-scale models at similar resolutions for decades.

*Also, a large part of the word is karstic, which is not reflected in the global geological model and conceptualization of MODFLOW. These areas should be fully blanked out, global maps of karst are available. Note that fractured systems are also treated the same as porous aquifers, there is no mention of the conceptual incompatibility of this approach. Moreover, all areas with permafrost or snow cannot be simulated either with the current conceptualization and should be blanked out as well.*

We will mask out the karst areas that are part of the alluvial sedimentary basins in the global maps. We will also indicate in the maps the areas that do not belong to the alluvial sedimentary basins, which include fractured systems. We do not agree with masking out the permafrost, as these areas are implicitly accounted for by the global hydrological model used to force the groundwater model (with recharge, groundwater withdrawal rates and surface water levels). In this model, permafrost is included by

impermeable soils that generate no groundwater recharge. Hence, we simulate no active groundwater flow systems below permafrost areas, only the presence of groundwater.

*The list could be endlessly expanded, and this assessment should have been done before the submission of the paper.*

We would like to stress again that we use a previously published global groundwater and explicitly focus on the computational challenge to increase its spatial resolution with a factor 100. This is one of the many steps needed to arrive at significantly better global groundwater models (see above). For a discussion on these steps and what could be done to achieve them we refer again to Condon et al. (2021) and Gleeson et al. (2021).

*The model only seems to report a groundwater decline.*

This is actually not the case. We truncated the legend to focus on decline only. But of course interannual to interdecadal climate variability could be an important cause of both decline as well as increase of groundwater levels. Therefore, in an updated version of Figure 14c, we will show both negative and positive trends.

*In Figure 14a there are no hydraulic heads above the surface. But this is the case for many confined aquifers, take the great artesian basin in Australia or the Nubian systems for example.*

Thanks for pointing this out. Again, this is not the case and is again caused by our truncating of the legends. Also, for regions with confined aquifers, we have portrayed the groundwater levels in the confining layer, not the underlying aquifer. These groundwater levels are kept at bay by drains positioned at surface level to emulate groundwater exfiltration for groundwater in undated areas. We will the legend to allow for groundwater levels and heads above the surface. Also, we will add a separate figure showing the heads in the confined aquifers. This shows several areas with heads above the surface. Of course, there are still areas where one would expect larger heads. This can be explained by the lack of a parameterization of mountain front recharge in the model, which is an important source of recharge of confined aquifers such as the Great Artesian basin, Australia.

*The large depths to groundwater in all mountainous regions further show that the model results have nothing to do with reality. You can easily see this by consulting the measured hydraulic heads as I suggested above, or by taking a healthy hike in the mountains and appreciating countless small rivers and streams emerging at high altitudes. Again, this list could be endlessly expanded.*

The mountain areas are also part of the evaluation shown in Figure 12. Here, errors are indeed large if groundwater levels are measured in mountain valleys that are too small to be resolved by the original 5 arcminute hydrogeology of De Graaf et al. (2017). Therefore, the model is only capable of simulating the mountain block hydraulic heads in the mountain areas and not the groundwater pockets in mountain valleys higher up. This limitation was already mentioned in De Graaf et al. (2015) but we agree we will repeat this here in the Introduction again. Also, we expect this to greatly improve if one re-parameterizes the hydrogeological model of De Graaf et al. (2017) at 30-arc-second resolution. Note this is work in progress but outside the scope of this paper.

*Little is said about the water balances of the catchments.*

[Figure]

The water balance is better evaluated at the aquifer level, since groundwater flows across the catchment boundaries. Water balances are by definition closed (apart from a minor numeric error) in our MODFLOW simulations since they are part of the closure criterion of the numerical solution.

Required conditions for publication

Reviewer 1 states a number of conditions that need to be met before the reviewer can support publication. We state these conditions in italics below and provide a response to what extent these conditions can be reasonably met given the scope of this paper.

- *Greatly expand the discussion on the fundamental conceptual issues and demonstrate to what extent they undermine the robustness of the model.*
- We will extent section 2.2.1 by stating the conceptual choices made in the original 5 arcminute version of the model which also pertain to this version. We will also state the limitations of this setup, i.e. the six limitations stated above. We are not sure what the reviewer means with effect on the robustness of the model, but it may be to show how well the model does in areas where it is not meant to work that well. In addition to the steady-state evaluation, we will add extra curves for the GLOBGM to Figure 12b showing the errors for sedimentary basins without karst, sedimentary basins and karst and non-sedimentary basins. *Blank out areas where karst and permafrost snow is present.*

We will provide a mask portraying the areas not belonging to the sedimentary basins and the karst areas in the sedimentary basins.

- *Demonstrate which fractures systems can be simulated with a Darcy type approach (some can but not all, depending on the properties). Blank out the ones that cannot*

With all due respect, we think that this request is not reasonable in the context of this paper. Demonstrating which can and cannot be simulated with Darcy-type flow, which also is dependent on the scale of analysis, will never be possible, even at the local scale. But since the non-sedimentary (crystalline) rocks will be masked out in a newer manuscript version, this is also not needed.

- *Refine the spatial resolution of the model that groundwater abstraction and rivers are at least conceptually implemented correction. With a finite difference approach as implemented here this is essentially impossible with the current computational capacities, so the grid should be fully redone using the unstructured features of MODFLOW which now unfortunately are only used to deal with issues in islands. Pumped aquifers or river have to be simulated with a spatial resolution adequate to the physics of the problem.*

We respectfully disagree that re-gridding is necessary. It is a misconception that one needs to refine the grid to a certain point to correctly capture the physics of the problem. The physics are resolved: Darcy's law and continuity of mass are obeyed at the scale of the numerical grid. Of course, to maintain scale-consistency, the right scaling laws of hydrogeological parameters are required and we have tried to do so as much as possible. Regular grids have been used many decades in regional-scale groundwater modelling, accepting that features smaller than the grid size are not resolved. Sub-grid parameterization or scaling laws then make sure that fluxes remain more or less constant with scale. For instance, sub-grid groundwater-surface water interaction of non-resolved streams can be taken care of by putting in additional drains (see e.g., De Graaf et al., 2017). Non-structured grids to resolve fine features of interest is something we are definitely thinking about implementing in the near future.

[Figure]

- *Demonstrate that the geological models and the soil maps for the remaining areas are an adequate simplification of the complexity of the subsurface. Demonstrate that populating the geological/ soil models with physical parameters is robust.*

With all due respect, we think that this is not a reasonable request. First, what is "adequate" here? Second, even if adequate could be defined, it would greatly depend on the purpose for which the model is used and at what scale (e.g. global sensitivity versus local prediction). Third, this would then mean finding alternative more detailed hydrogeological models, e.g. as used in regional groundwater models, inserting these into our model and look at the differences.  This is indeed an interesting exercise, but an enormous task reserved for some future research out of the scope of this paper.

- *Test the model with real data, including streamflow, and demonstrate that the workarounds in section 2.4.2 are robust.*

The section on model evaluation 3.3 we do compare the model with real mean head data and time series over 900k locations in the US. Comparison with streamflow data does not make sense, since the groundwater model only produces baseflow. However, we could compare the streamflow data of the land surface with discharge observations (theoretically land surface baseflow and groundwater baseflow are on average the same). However, this has been done extensively in Sutanudjaja et al. (2018). We refer to that paper for these specific validation statistics.

- *The outcome of such an assessment will show that the model is far away from being used in any real-world context. The paper needs a clear statement in the abstract and the conclusions on this.*

As already explained under point 2 in the beginning of this reply: if models are inaccurate in certain areas, they can still be useful for (sub-)regional sensitivity studies of global extent as some of the references given show. We therefore feel that this model is far away from being used *in any context* is too strong. However, we will provide a caveat in the text related to the limitations of global groundwater models, when we describe the model in section 2.2.1.

References

Alcamo, J., P. Döll, F. Kaspar, and S. Siebert (1997), Global change and global scenarios of water use and availability: An application of WaterGAP 1.0., Rep. A9701, Cent. for Environ. Syst. Res., Univ. of Kassel, Kassel, Germany.

Condon, L. E., Kollet, S., Bierkens, M. F. P., Fogg, G. E., Maxwell, R. M., Hill, M. C., et al. (2021). Global groundwater modeling and monitoring: Opportunities and challenges. Water Resources Research, 57, e2020WR029500.

De Graaf, I. E. M., Sutanudjaja, E. H., van Beek, L. P. H., & Bierkens, M. F. P. (2015). A high-resolution global-scale groundwater model. Hy-drology and Earth System Sciences, 19(2), 823–837.

De Graaf, I. E. M., van Beek, R. L. P. H., Gleeson, T., Moosdorf, N., Schmitz, O., Sutanudjaja, E. H., & Bierkens, M. F. P. (2017). A global-scale two-layer transient groundwater model: Development and application to groundwater depletion. Advances in Water Resources, 102, 53–67.

[Figure]

De Graaf, I. E. M., Gleeson, T., van Beek, L. P. H., Sutanudjaja, E. H., & Bierkens, M. F. P. (2019). Environmental flow limits to global ground-water pumping. Nature, 574(7776), 90–94.

Fan, Y., Li, H. & Miguez-Macho, G. Global patterns of groundwater table depth. Science 339, 940–943 (2013).

Fan, Y., Miguez-Macho, G., Jobbágy, E. G., Jackson, R. B., & Otero-Casal, C. (2017). Hydrologic regulation of plant rooting depth. Proceedings of the National Academy of Sciences of the United States of America, 114(40), 10572– 10577.

Gleeson, T., Wagener, T., Döll, P., Zipper, S. C., West, C., Wada, Y., Taylor, R., Scanlon, B., Rosolem, R., Rahman, S., Oshinlaja, N., Maxwell, R., Lo, M.-H., Kim, H., Hill, M., Hartmann, A., Fogg, G., Famiglietti, J.S., Ducharne, A., de Graaf, I., Cuthbert, M., Condon, L., Bresciani, E., and Bierkens, M.F.P. (2018), GMD perspective: The quest to improve the evaluation of groundwater representation in continental- to global-scale models, Geosci. Model Dev., 14, 7545–7571.

Manabe, S. (1969), Climate and the ocean circulation: The atmospheric circulation and the hydrology of the earth's surface, Mon. Weather Rev., 97, 739–774.

Miguez-Macho, G., and Y. Fan (2012), The role of groundwater in the Amazon water cycle: 2. Influence on seasonalsoil moisture and evapotranspiration,J. Geophys. Res.,117, D15114.

Minderhoud, P.S.J., Erkens, G., Pham, V.H., Bui, V.T., Erban, L.E., Kooi, H., Stouthamer, E., 2017. Impacts of 25 years of groundwater extraction on subsidence in the Mekong delta, Vietnam. Environ. Res. Lett. 12, 13.

Prentice, I. C., W. Cramer, S. P. Harrison, R. Leemans, R. A. Monserud, and A. M. Solomon (1992), A global biome model based on plant physiology and dominance, soil properties and climate, J. Biogeogr., 19, 117–134.

Reinecke, R., Foglia, L., Mehl, S., Trautmann, T., Cáceres, D., & Döll, P. (2019). Challenges in developing a global gradient-based groundwater model (G3M v1.0) for the integration into a global hydrological model. Geoscientific Model Development, 12(6), 2401–2418.

Sutanudjaja, E.H., van Beek, R., Wanders, N., Wada, Y., Bosmans, J.H.C., Drost, N., van der Ent, R J., de Graaf, I.E.M., Hoch, J.M., de Jong, K., Karssenberg, D., López López, P., Peßenteiner, S., Schmitz, O., Straatsma, M.W., Vannametee, E., Wisser, D. and Bierkens, M. F. P. (2018), PCR-GLOBWB 2: a 5 arcmin global hydrological and water resources model, Geosci. Model Dev., 11, 2429–2453.

**Reply to comment of Reviewer 2 om GMD-2022-226**

General reply

We thank the reviewer for taking the time to evaluate our manuscript and the extensive comments. Below we present a point-by-point reply to the points raised by Reviewer 2. The Comments of the Reviewer are represented in Italics, our replies in Roman.

GENERAL COMMENTS:

[Figure]

*The manuscript is well-organized and successfully demonstrates a computational workflow towards the efficient simulation of long term monthly GW flow and discharge at ~ 1 km scales at a global extent. For this reason it is an important contribution.*

We thank Reviewer 2 for this positive evaluation of the contribution.

*My general concern with the manuscript is that the (i) benefit of increasing the model resolution and (ii) the resulting utility (or persisting limitations) of a 30″ GW simulation are assumed rather than clearly motivated. While the introduction (55ff) names the need for better GW estimates and describes the opportunity afforded to GW modelers by datasets generated at increasing resolution, the manuscript would be strengthened by persuading the reader that (a) 'better GW estimates' need to be made by a global model; (b) a 30″ model makes better estimates than a 5′ model for some set of GW questions; and/or (c) the development of a 30″ model is a provisional but critical step towards the type of better, global GW estimates that are at some point in the future. Some discussion of these factors would also better situate the discussion of model performance in Section 3. While it is true that the paper is about the technical dimensions of model construction and execution, and a full interrogation of model performance may belong in an altogether separate paper, models must be suitable for some purpose(s), and it is not discussed whether this model is or is not, or is on the way to being so.*

These are valid points by the reviewer and these have also been indirectly raised by Reviewer 1.

When it comes to the critical steps that are needed to obtain better global GW estimates, or better global groundwater models in general, we will now add a section to the Introduction that states (based on other literature) which developments are needed to improve global groundwater models beyond the current state-of-the-art. This will be a statement that reads something like (see also the reply to Reviewer 1):

"When it comes to the steps needed to make a leap change in improving the current generation of models, many steps are still needed; see also the reviews by Condon et al. (2021) and Gleeson et al. (2021). Specifically these are: 1) improved hydrogeological schematization, particularly including multilayer semi-confined aquifer systems and the macroscale hydraulic properties of Karst and Fractured systems; 2) increased resolution to better resolve topography and in particular resolve smaller higher altitude groundwater bodies in mountain valleys; 3) improved knowledge on location, depth and rate over time of groundwater extractions; 4) better estimated groundwater recharge, especially in drylands and at mountain margins; 5) increased computational capabilities to be able to make simulations with the above improvements possible. Our paper specifically revolves around items 2 and 5: *If we improve spatial resolution, how should we make this computationally possible?* It is a small but important step to better global groundwater that needs to be taken to proceed further."

Regarding the (persisting) limitations that even a 30 arcsecond model still suffers, we intend to add a paragraph to the beginning of section 2.2.1 highlighting the limitations the global groundwater model (as it derives from de model of De Graaf et al. 2015; 2017) has in simulating global groundwater levels and heads (see also the reply to Reviewer 1):

Particularly we will state that: "The application domain of the model is as follows: 1) it is intended to simulate hydraulic heads in the top aquifer systems, so unconfined aquifers and the uppermost confined aquifers; 2) wherever there are multiple stacked aquifer systems, these are simplified in the model to one confining layer and one aquifer; 3) the model schematization is suitable for heads in large sedimentary alluvial basins (main productive aquifers) that have been mapped at a 5-arcminute resolution; 4) in as far

these sedimentary basins include karst, it is questionable of a Darcy approach can be used to simulate large-scale head distributions; 5) due to the limited resolution of the hydrogeological schematizations, in mountain areas we simulate the heads in the mountain blocks but not those of groundwater bodies in hillslopes and smaller alluvial mountain valleys; 6) also, for the heads in the mountain blocks, we assume that secondary permeability of fractured hard rock can also simulated with Darcy groundwater flow; an assumption that may be questioned."

*Section 3 should briefly help the reader understand that. For example, the steady state Fan (2017) model, which is at the same 30" resolution as GLOBGM, performs better with respect to DTW observations; hypotheses about why this is so (conceptual model? recharge formulation? subsurface parameterization using newly available datasets?) would help guide subsequent GLOBGM development and analysis, and would help the reader understand the significance of the model comparisons included in this manuscript.*

We will add a description to Section 3.3.1 about the presumed reasons for the improvement performance of the 30 arcsecond model compared to the 5-arcminute model. We think that the main reason is that the higher resolutions are better in following topography and relief, in particular to resolve smaller higher altitude groundwater bodies in mountain valleys. Also, higher resolution models have a smaller scale gap with the in-situ head observations in wells. The reason that the Fan et al. (2013) model does better is that their model has been calibrated, while ours has not. They do this by optimizing a two-parameter relationship between the e-folding depth value that reduces conductivity with depth and the topographic slope. So, they effectively calibrate the transmissivity of their model. For the US, many of the groundwater level observations they use for this calibration are also part of the validation dataset. We will include this explanation in Section 3.3.1 as well.

*SPECIFIC COMMENTS:*

*121ff (and Figure 2). I do not understand how 'sorting by cell count' constrains or informs the decomposition of the global domain into 9050 separate 'grids'.*

The sorting of the grids using the cell count is important to determine the largest models subject to parallelization. After sorting from large to small, the first three largest directly correspond to the grids for the Afro-Eurasia, Americas and Australia models, respectively. We will add some text explaining the purpose of sorting.

*Figure 2 should note that, while helpful for illustrative purposes, this submodel distribution does not actually occur in the global domain decomposition because in GLOBGM all of the islands belong to a single model.*

Although Figure 2 is indeed for illustrative purposes, the submodel distribution illustrated in this figure actually does occur in the global domain decomposition: the largest island in model 1 on the left corresponds to the Afro-Eurasia, Americas or Australia model; the three islands in model 2 on the right correspond to the Islands model. Although the Islands model contains 9047 islands, the larger "islands" may be split up to run on multiple cores. This is actually the case for the largest "island" Great Britain in the model when using 32 cores as in Table 4. We will add text to the paragraph describing Figure 2, line 189, to improve the connection with the global domain decomposition.

*Figure 3: is the Node Selection Workflow before the Model Workflow? I'm not clear on the logical or temporal relationship between the 2 workflows.*

As shown in Figure 3b, the Node Selection Workflow makes usage of the Model Workflow, see line 293 "Then, the Model Workflow generates the model input files for this number of cores". We will emphasize this relation, directly at the beginning of the Node Selection workflow description, line 288, as well as in the caption of Figure 3b.

*Section 3.1.2 - How does tiling reduce storage requirements?*

This is described in Section 2.3.2, line 306, "using tiles cancels a significant number of redundant sea cells (missing values)". Although in section 3.1.2 we refer to this in line 469 "Using data tiles therefore saves storing ~8 TB of redundant data (43% reduction; see Section 2.3.2)", we will clarify this more in the text of 2.3.2. Basically, for the global extent, we use 163 square tiles instead of 288 tiles (24 x 12; including all oceans, Greenland and Antarctica that are not used as input for our model), and therefore for each global map we only need to store 100 x 163/288 = 57% of the data corresponding to 43% reduction. Hence, instead of storing one 30-arcsec global raster file having 43200x21600 cells, we store 163 smaller raster files, each having 1800x1800 cells.

*TYPOGRAPHIC/SYNTACTICAL SUGGESTIONS AND CORRECTIONS*

Thank you for the close reading. We will include these suggestions and corrections in our revised manuscript.

References

Condon, L. E., Kollet, S., Bierkens, M. F. P., Fogg, G. E., Maxwell, R. M., Hill, M. C., et al. (2021). Global groundwater modeling and monitoring: Opportunities and challenges. Water Resources Research, 57, e2020WR029500.

De Graaf, I. E. M., Sutanudjaja, E. H., van Beek, L. P. H., & Bierkens, M. F. P. (2015). A high-resolution global-scale groundwater model. Hy-drology and Earth System Sciences, 19(2), 823–837.

De Graaf, I. E. M., van Beek, R. L. P. H., Gleeson, T., Moosdorf, N., Schmitz, O., Sutanudjaja, E. H., & Bierkens, M. F. P. (2017). A global-scale two-layer transient groundwater model: Development and application to groundwater depletion. Advances in Water Resources, 102, 53–67.

Fan, Y., Li, H. & Miguez-Macho, G. Global patterns of groundwater table depth. Science 339, 940–943 (2013).

Gleeson, T., Wagener, T., Döll, P., Zipper, S. C., West, C., Wada, Y., Taylor, R., Scanlon, B., Rosolem, R., Rahman, S., Oshinlaja, N., Maxwell, R., Lo, M.-H., Kim, H., Hill, M., Hartmann, A., Fogg, G., Famiglietti, J.S., Ducharne, A., de Graaf, I., Cuthbert, M., Condon, L., Bresciani, E., and Bierkens, M.F.P. (2018), GMD perspective: The quest to improve the evaluation of groundwater representation in continental- to global-scale models, Geosci. Model Dev., 14, 7545–7571.

---

## Author Response (AR1)

To: Editors of Geoscientific Model Development

**Faculty of Geosciences**
Department of Physical Geography
PO Box 80.115, 3508 TC Utrecht, The
Netherlands

**Visiting address**
Vening Meinesz A, Princetonlaan 8a
3584 CB, Utrecht, The Netherlands

**Website**
www.uu.nl/geo/fg
**Email** jarno.verkaik@deltares.nl
**Phone** +31(0)6 4691 4636

**Date: May 31, 2023**
**Subject: revised manuscript**

Dear Editor, Dear Wolfgang Kurtz,

We have improved our manuscript with all the reviewers' comments, and we are happy to inform you that we have managed to add a comparison with GRACE data. We have implemented the changes as we suggested in our final response of April 19, 2023. Please see in red below the specific lines of the changes in the revised document:

1. *Introduction:* Based on literature, we will add a paragraph to the beginning of Section 1 about which developments are needed to improve global groundwater models beyond the current state-of-the-art. This will also identify as one these improvements increasing the resolution and increasing the speed of computation to make this possible. This will then immediately make clear which of these possible improvements are the focus of this paper.
   See line 51-60 'From the work... proceed further.', and line 65-67 'and the native... GLOBGM'.

2. *Methods:* In Section 2.2.1, "Description of the GLOBGM", we will add a paragraph about the assumptions and limitations of our GLOBGM model with respect the work for the 5′ global groundwater model, De Graaf et al. (2015, 2017). In this, we will again be more explicit that our work addresses *the computational challenges need to be overcome if the resolution of this model is increased.*
   See line 248-258 'The application... questioned'.

3. *Steady-state model evaluation:*

   3.1. We will add extra curves for our GLOBGM to Figure 12b showing the errors for sedimentary basins without karst, sedimentary basins and karst and non-sedimentary basins.
   For this purpose, we added Figure A1 and put this in the Appendix.

   3.2. We will mask out the karst areas in the steady-state global maps in Figure 14a.
   Change made to Figure 15a (former Figure 14a).

   3.3. We will change the legend of in Figure 14a to allow for groundwater levels and heads above the surface.
   Change made to the legend of Figure 15a.

   3.4. We will add a separate figure to Figure 14 showing the heads in the confined aquifers, and mask out the Karst areas.
   For this purpose, we added Figure A2 to the Appendix.

4. *Transient model evaluation:*

[Figure]

4.1. We will mask out the karst areas for the global maps of Figure 14b and Figure 14c.
Change made for Figure 15b and Figure 15c (former Figure 14b and Figure 14c).

4.2. We will change the legend of Figure 14c for showing both positive and negative trends.
Change made to Figure 15c.

4.3. We will add a supplemental document explaining the filtering of transient well locations in more detail.
We added a new document to the supplemental.

5. We will investigate the possibilities for comparing with GRACE data for this version of our GLOBGM, version 1.0, or an upcoming version.
We added the GRACE analysis to our manuscript. See section 2.4.3 and 3.3.3, Figure 14, and conclusion line 702 'Monthly and multi-year…'.

6. We will add text to Section 2.1 for better explaining the purpose of sorting for the global domain decomposition.
See line 128-130, 'Furthermore… require parallelization'.
Furthermore, in this Section, we will add text to improve the connection of the example of Figure 2 to the actual global domain decomposition being used.
See line 213-216, 'The outline situation… Australia model'.

7. In Section 2.3.1, we will add text to emphasize better the relationship between the "Model Workflow" and the "Node Selection Workflow".
See line 317-318 'Figure 3b… model i'.

8. We will include the typographical/syntactical suggestions and corrections.
These minor changes were made throughout the entire manuscript.

We hope that our changes in our manuscript are adequate and will provide a good starting point for a third reviewer. Please let me know if you have any questions or more clarification is required.

On behalf of all authors,

With kind regards,

Jarno Verkaik

---

## Referee Report (RR1)

I have reviewed the manuscript "*GLOBGM v1.0: a parallel implementation of a 30 arcsec PCRGLOBWB- MODFLOW global-scale groundwater model*" by Verkaik et al. They use a global-scale GGC (30") physics-based groundwater flow model with a parallel implementation in order to reduce model run times.

The main objective of the work is clearly stated. For evident numerical reasons, such global-scale model must have to take the most from parallel computing to reach "*acceptable*" simulation runtimes.

The text is mostly readable by the wider groundwater community (which I think is the intended readership for this paper), although I would suggest adding some more detail and/or vulgarising some analytical "jargon" such as the so-called "Pfafsetter level". This makes the methods section a bit arched for groundwater scientists who are not specialised in computational techniques. However, this is not a critical point for general understanding.

I do not have technical comments that would tell me to reject this paper. My only major comment is rather on the over-consideration in the precision of such models to simulate arbitrary predictions at regional to local scale (for surface water capture for instance). What are the general benefits to our community of having these global scale models? To put it more clearly, what is the main objective of developing such a model?

Although it is evident for atmospheric sciences that global scale physics-based models have to deal with processes such as la Niña or el Niño; encompassing physical terrestrial boundaries; it is not so clear to me what would be the benefits of groundwater modelling at global scale. Why do we need to overcome the physical boundaries that define independent hydrogeological systems? To what extent can such a global scale model perform better than a regional scale model specifically developed for a local hydrogeological system? Finally, what is the main purpose (i.e. prediction) of such models? The paper would benefit from including such clarifications in the introduction.

Despite the obvious limitations that I have discussed previously, why do you not consider other type of observations to "validate" the model, such as stream flow data and/or satellite based data?

Other comments

In lines 43 to 45 you justify the need to assess groundwater depletion, but why do we need global scale models for this? Would regional scale models be more appropriate instead?

Line 48: So this is a two layer model. This is very coarse for the vertical direction. Why not consider a 2D approach? At this scale I do not see the benefit of including the 3D at this coarse resolution. Perhaps more explanation is needed here (without having to search for the information in the many papers you refer to).

Line 75, more detail is needed here. The reference to Gleeson et al, 2021 is not sufficient.

Line 99: Why is the upper model layer a confined layer? I would rather conceptualize the upper layer as an "unconfined" layer and the bottom layer as a "confined" layer. Not clear.

Line 103-104: Why is the water not allowed to leave the domain at the upper layer? How do the model deal with seepage faces/nodes? Is water can leave the model domain from other boundary conditions than rivers and lakes? The conceptualization of model boundary conditions is not very clear.

Line 421: The average amplitude error is not so straightforward. Why not just consider the residuals, which is more often used in groundwater modelling applications?

Line 505: typo: "het"

Line 568: typo: "be left"

Section 3.3: A CONUS-extent (US) is considered to validate the global scale model. Model validation is therefore conducted on a smaller scale than the global-scale. It seams to be "cherry-picked' to favour a "region" where the model is better constrained. This is where satellite based data can be useful for instance to validate over the globe.

Figure 12: It seems that the GIM model of Fan et al., 2017 performs better than the current physics-based MODFLOW model with 30" resolution. How do you explain this? Although the model of Fan et al., 2017 can be calibrated, can such a model be 'calibrated' using any of the currently available methods? If so, calibrated for what? Heads? Flow? Model calibration must be carried out with the aim of reducing the uncertainty of a given prediction. This is where the definition of the purpose of the model is very important.

Line 632: Is this type of model really intended for the "average user"?

Line 643: It could be dangerous to reduce the number of model iterations, as this is likely to increase numerical errors. I would not advise this, especially for large models where small numerical errors can lead to large errors in the fluxes.

Line 649: So this (i.e. the memory limitation) completely precludes the use of sophisticated inverse modelling and uncertainty analysis. On the one hand you reduce the run time of the forward model, but on the other hand you increase the memory requirement. This looks like an intractable problem.

---

## Author Response (AR2)

To: Editors of Geoscientific Model Development

Faculty of Geosciences
Department of Physical Geography
PO Box 80.115, 3508 TC Utrecht, The Netherlands

Visiting address
Vening Meinesz A, Princetonlaan 8a
3584 CB, Utrecht, The Netherlands

Website
www.uu.nl/geo/fg
Email jarno.verkaik@deltares.nl
Phone +31(0)6 4691 4636

Date:  September 21, 2023
Subject: second revised manuscript

Dear Editor, Dear Dr. Wolfgang Kurtz,

First, we thank the reviewers for their time and efforts to further increase the quality of our manuscript. We have taken account of all the reviewers' comments. Regarding Reviewer #2, we implemented all the (minor) changes that he / she suggested. Regarding Reviewer #3 and #4, please find below the comments of the reviewers in *italics* and our replies in roman.

Reviewer #3

GENERAL COMMENTS

*I have reviewed the manuscript "GLOBGM v1.0: a parallel implementation of a 30 arcsec PCRGLOBWB-MODFLOW global-scale groundwater model" by Verkaik et al. They use a global-scale GGC (30") physics-based groundwater flow model with a parallel implementation in order to reduce model run times.*

*The main objective of the work is clearly stated. For evident numerical reasons, such global-scale model must have to take the most from parallel computing to reach "acceptable" simulation runtimes.*

*The text is mostly readable by the wider groundwater community (which I think is the intended readership for this paper), although I would suggest adding some more detail and/or vulgarising some analytical "jargon" such as the so-called "Pfafsetter level". This makes the methods section a bit arched for groundwater scientists who are not specialised in computational techniques. However, this is not a critical point for general understanding.*

We thank Reviewer #3 for this brief overview of our paper. We agree to make our manuscript readable a wide groundwater community and thus have performed another round of edits in which we tried to add some additional explanations for some technical terms, such as Pfasfstetter level (e.g., line 237):

HydroBASINS catchments follow the Pfafstetter base-10 coding system for hydrologically coding river basins, where the main stem is defined as the path which drains the greatest area, and at each refinement level ten areas are defined: four major tributaries, five inter-basin regions and one closed drainage system (Verdin and Verdin, 1999).

*I do not have technical comments that would tell me to reject this paper. My only major comment is rather on the over-consideration in the precision of such models to simulate arbitrary predictions at regional to local scale (for surface water capture for instance). What are the general benefits to our community of having these global scale models? To put it more clearly, what is the main objective of developing such a model?*

*Although it is evident for atmospheric sciences that global scale physics-based models have to deal with processes such as la Niña or el Niño; encompassing physical terrestrial boundaries; it is not so clear to me what would be the benefits of groundwater modelling at global scale. Why do we need to overcome the physical boundaries that define independent hydrogeological systems? To what extent can such a global scale model perform better than a regional scale model specifically developed for a local hydrogeological system? Finally, what is the main purpose (i.e. prediction) of such models? The paper would benefit from including such clarifications in the introduction.*

This point has been made many times before and can be made for all global impact models that are out there, such as global hydrological models (used for global water resources assessments), global dynamic vegetation models and agricultural models (used for global terrestrial carbon and biodiversity and food security assessments), global economic models (to assess global economic impacts) and global inundation models (used to assess global flood risk). All of these global assessments could also be made with a patchwork of regional-scale models. Still those models exist for global scale assessments, sometimes standalone, sometimes as part of integrated assessment models or earth system models. The reason is that a complete coverage with regional-scale models is generally not available, while a global model has the benefit of uniformity of model setup such that it is more straightforward to compare different regions in the world.  This is especially useful in data-scarce areas where often no models are available. With this global approach, one has at least a first order model, which incorporates hydrogeological concepts of areas where one has more information. Apart from this, there are reasons to have a global groundwater model in global scale analysis that require a global coverage. A first example is terrestrial vegetation growth and evaporation that are modulated by the presence of shallow groundwater tables with possibly global-scale impacts on the global water and carbon cycle under climate change (Anyah et al., 2008; Miguez-Macho and Fan, 2012). Another example is the importance of non-renewable groundwater use to global food production and food trade (Dalin et al., 2017) with possible global impacts on future food security. A third example is contribution of terrestrial water storage change on regional sea-level trends (Karabil et al., 2021).  In all of these cases the complete global groundwater system has to be considered.

To make this case a bit stronger we have added the following lines to the Introduction (line 51):

Recent publications have called for a better representation of groundwater in earth system models (Bierkens et al., 2015; Clark et al., 2015; Gleeson et al., 2021). Apart from providing a globally consistent and physically plausible representation of groundwater flow using a uniform model set-up, global-scale groundwater models could serve to support global change assessments that depend on a global representation of groundwater resources. Examples of such assessments are the impact of climate change on vegetation, evaporation and atmospheric feedbacks (Anyah et al., 2008; Miguez-Macho and Fan, 2012), the role of groundwater depletion in securing global food security and trade (Dalin et al., 2017) and the contribution of terrestrial water storage change to regional sea-level trends (Karabil et al., 2021).

*Despite the obvious limitations that I have discussed previously, why do you not consider other type of observations to "validate" the model, such as stream flow data and/or satellite based data?*

[Figure]

These are good suggestions which we have partly followed in the second version of our paper (which we hope Reviewer #3 has been evaluating), see e.g. Section 2.4.3 and Section 3.3.3. We have contemplated using streamflow data, but we are only presenting the groundwater component of the terrestrial hydrological cycle, so we only simulate groundwater discharge with the groundwater model, which cannot be directly compared with total streamflow observations. We have compared total water storage anomalies based on our head simulations and other storage components from the driving hydrological model PCR-GLOBWB 2 (Sutanudjaja et al., 2018) with observations from GRACE and GRACE-FO (Figure 15, line 723) with good results.

OTHER COMMENTS

Judging from the line numbers mentioned, we are afraid that the Reviewer #3 has evaluated the original manuscript and not the revised second version that we submitted after a first round of revisions. So, to answer the questions below we have also included the original first version of the manuscript in our answers.

*In lines 43 to 45 you justify the need to assess groundwater depletion, but why do we need global scale models for this? Would regional scale models be more appropriate instead?*

Here, we refer to our answer above. Regional models could be used for that in principle, but there would be no global coverage and no spatially consistent and uniform way of determining groundwater depletion.

*Line 48: So this is a two layer model. This is very coarse for the vertical direction. Why not consider a 2D approach? At this scale I do not see the benefit of including the 3D at this coarse resolution. Perhaps more explanation is needed here (without having to search for the information in the many papers you refer to).*

To be fair, we do not mention three-dimensional or 3D anywhere in the paper. We have in fact a quasi-3D model where we simulate 2D horizontal flow in an unconfined aquifer or a confined aquifer topped with a confining layer, with vertical groundwater exchange between the confined aquifer and the overlying confining layer. However, the model could be extended to a fully 3D modelling approach if wanted and when more information about the vertical geological layering is available at the global scale. Note this is work for the future.

*Line 75, more detail is needed here. The reference to Gleeson et al, 2021 is not sufficient.*

We have changed this sentence to (line 96):

We provide a limited evaluation of the computed results, and we note that the current model is a first version that should be further improved in the future. We refer to Gleeson et al. (2021) for an extensive discussion on pathways to further evaluate and improve global groundwater models.

*Line 99: Why is the upper model layer a confined layer? I would rather conceptualize the upper layer as an "unconfined" layer and the bottom layer as a "confined" layer. Not clear.*

Sorry for being unclear here. But we talk about a confining layer, not a confined layer. A confining layer is a less permeable layer on top of a confined aquifer. Flow is mostly vertical in a confining layer. Where the confining layer is present, we have a confined aquifer below and where it is not present the second layer is an unconfined aquifer.

To make this clearer we changed the text as (line 124):

*The 5' PCR-GLOBWB-MODFLOW global-scale groundwater model (GGM) consists of two model layers: where a confining layer (having a lower permeability) is present, the upper model layer represents the confining layer and the lower layer a confined aquifer. If a confining layer is not present, both the upper and lower model layers are part of the same unconfined aquifer (de Graaf et al., 2015, 2017).*

*Line 103-104: Why is the water not allowed to leave the domain at the upper layer? How do the model deal with seepage faces/nodes? Is water can leave the model domain from other boundary conditions than rivers and lakes? The conceptualization of model boundary conditions is not very clear.*

We acknowledge that this is indeed not clear. We have added (line 189):

*Note that in the GLOBGM interaction with surface water or surface drainage is modelled by putting rivers and drains in the first active layer, seen from top to bottom.*

*Line 421: The average amplitude error is not so straightforward. Why not just consider the residuals, which is more often used in groundwater modelling applications?*

Note we implicitly do consider the residuals, but this is done by comparing: 1) the average error (bias) by evaluating the steady state model results with observed average groundwater depths; 2) looking at the timing of peaks and throughs by calculating the cross-correlation between observations and simulations. The amplitude error is additionally calculated to see if the variation is represented correctly. This has been done in previous work (De Graaf et al., 2017) and makes it comparable to that work. The three components, viz. bias, timing error and amplitude error are also the three aspects of e.g., the Kling-Gupta Efficiency (KGE) which is also often used to evaluate the fit to time series. We additionally added a comparison of the trend. We believe that this still provides a comprehensive way to evaluate the performance of a groundwater model.

*Line 505: typo: "het"*
*Line 568: typo: "be left"*

Thanks for noticing. We have corrected this.

*Section 3.3: A CONUS-extent (US) is considered to validate the global scale model. Model validation is therefore conducted on a smaller scale than the global-scale. It seams to be "cherry-picked' to favour a "region" where the model is better constrained. This is where satellite based data can be useful for instance to validate over the globe.*

This is exactly what we did. In the first version of the manuscript that Reviewer #3 is referring to, we restricted the validation to the U.S. because that is the continent with readily available (open) groundwater level data and with multiple other models in place. However, in the second version we extended the evaluation to the globe by comparing results with GRACE and GRACE-FO satellite-based gravity anomaly data (See Section 2.4.3 and 3.3.3 and Figure 13 / Figure 14 is second revised manuscript).

*Figure 12: It seems that the GIM model of Fan et al., 2017 performs better than the current physics-based MODFLOW model with 30" resolution. How do you explain this? Although the model of Fan et al., 2017 can be calibrated, can such a model be 'calibrated' using any of the currently available methods? If so,*

[Figure]

*calibrated for what? Heads? Flow? Model calibration must be carried out with the aim of reducing the uncertainty of a given prediction. This is where the definition of the purpose of the model is very important.*

Both GIM and CGM have been calibrated by comparing simulated heads with observed head data and adjusting parameters. For GIM, this was done by iteratively changing the way hydraulic conductivity reduces with depth (thus in fact the transmissivity) till the simulated heads resemble the observed ones as good as possible. No such calibration has been performed with our model. This is left as work for the future, as the focus of this work was to introduce the numerical scheme that allows parallel computing to solve transient high-resolution multi-layer groundwater problems.

*Line 632: Is this type of model really intended for the "average user"?*

This is a fair point. We have removed "average" and just speak of users without access to large numbers of nodes.

*Line 643: It could be dangerous to reduce the number of model iterations, as this is likely to increase numerical errors. I would not advise this, especially for large models where small numerical errors can lead to large errors in the fluxes.*

We fully agree that numerical accuracy should not be lowered to prevent numerical errors. Our text was unfortunately unclear and here our intention was to refer to the parallel preconditioner in the linear solver that could be improved in the future to reduce the number of linear iterations without loss of any accuracy. Therefore, we changed the text as (line 786):

improving the parallel preconditioner for the linear solver to account for the increasing number of iterations

*Line 649: So this (i.e. the memory limitation) completely precludes the use of sophisticated inverse modelling and uncertainty analysis. On the one hand you reduce the run time of the forward model, but on the other hand you increase the memory requirement. This looks like an intractable problem.*

Note that we are here talking about disk storage, taken by input files and output files if all output is stored after each time step, and not about Random Access Memory (RAM) that is used by processors on nodes during computation. We don't expect that disk storage limitations precludes any sophisticated inverse modeling and/or uncertainty analysis. All the required input data need to be read in once and, as we show here, fit the processor memory of the collective nodes. Then multiple runs can be done as part of an uncertainty analysis or as iterations in an inverse scheme while parameters are updated in processor memory, as well as comparing parts of the outputs with observations that are also read in once. The feasibility of the uncertainty analysis or an inverse scheme thus depends only on the number of model runs needed. This could be speeded up by increasing the number of nodes used and is not limited by RAM requirements.

REFERENCES

*Anyah, R. O., Weaver, C.P., Miguez-Macho, G., Fan, Y. and Robock, A. (2008). Incorporating water table dynamics inclimate modeling: 3. Simulated groundwater influence on coupled land-atmosphere variability,J. Geophys. Res.,113, D07103, doi:10.1029/2007JD009087.*

[Figure]

*Bierkens, M. F. P. (2015). Global hydrology 2015: State, trends, and directions. Water Resources Research, 51(7), 4923–4947. https://doi.org/10.1002/2015wr017173*

*Clark, M. P., Fan, Y., Lawrence, D. M., Adam, J. C., Bolster, D., Gochis, D. J., et al. (2015). Improving the representation of hydrologic processes in Earth System Models. Water Resources Research, 51(8), 5929–5956. https://doi.org/10.1002/2015WR017096*

*Dalin, C., Wada, Y., Kastner, T. and Puma, M.J. (2017). Groundwater depletion embedded in international food trade. Nature 543, 700–704. https://doi.org/10.1038/nature21403*

*Gleeson, T., Wagener, T., Döll, P., Zipper, S. C., West, C., Wada, Y., Taylor, R., Scanlon, B., Rosolem, R., Rahman, S., Oshinlaja, N., Maxwell, R., Lo, M.-H., Kim, H., Hill, M., Hartmann, A., Fogg, G., Famiglietti, J. S., Ducharne, A., de Graaf, I., Cuthbert, M., Condon, L., Bresciani, E., and Bierkens, M. F. P. (2021) GMD perspective: The quest to improve the evaluation of groundwater representation in continental- to global-scale models, Geosci. Model Dev., 14, 7545–7571, 28. https://doi.org/10.5194/gmd-14-7545-2021, 2021*

*Karabil, S., Sutanudjaja, E.H., Lambert, E., Bierkens, M.F.P. and van der Wal, R. (2021). Contribution of land water storage change to regional sea-level rise over the twenty-first century. Frontiers in Earth Science, 9, 2021. https://doi.org/10.3389/feart.2021.627648*

*Miguez-Macho, G., and Fan, Y. (2012). The role of groundwater in the Amazon water cycle: 2. Influence on seasonalsoil moisture and evapotranspiration, J. Geophys. Res.,117, D15114, doi:10.1029/2012JD017540*

*Sutanudjaja, E. H., Van Beek, R., Wanders, N., Wada, Y., Bosmans, J. H. C., Drost, N., Van Der Ent, R. J., De Graaf, I. E. M., Hoch, J. M., De Jong, K., Karssenberg, D., López López, P., Peßenteiner, S., Schmitz, O., Straatsma, M. W.,Vannametee, E., Wisser, D., and Bierkens, M. F. P.: PCR-GLOBWB 2 (2018), A 5 arcmin global hydrological and water resources 850 model, Geosci. Model Dev., 11, 2429–2453, https://doi.org/10.5194/gmd-11-2429-2018.*

Reviewer #4

GENERAL COMMENTS

*Verkaik et al. present a study on the parallel setup and application of GLOBGM-MODFLOW at the global scale. They describe the parallel methodology and application including evaluation with observations over a limited area i.e CONUS. Overall, the manuscript is difficult to follow. It reads as if no iteration has been done on clarity and structure. Here, I am focusing on the (parallel) setup of the model and performance.*

We are sorry to read that the reviewer found it difficult to follow the manuscript. Since the reviewer is focusing on the parallel setup of the model and performance, we presume that the reviewer specifically has difficulties with Section 2.1. Therefore, we extensively restructured this section, renamed this to Parallelization approach, and introduced four (sub)sections labeled as: General concept (2.1.1); Procedure for deriving the independent unstructured grids (2.1.2); Defining the four groundwater models of the GLOBGM (2.1.3); Groundwater model partitioning: grid cell partitioning and catchment partitioning (2.1.4); and Node selection procedure (2.1.5). Section 2.1.1 summarizes the followed parallelization concepts, referring to the successive sections for details, and the general concept is now illustrated by a new figure, Figure 1. Where possible, we reused existing text and added new text mainly for Section 2.1.1 and Section 2.2.2. To improve the readability, we sharpened our definitions, e.g., in the revised manuscript we changed

the term "straightforward METIS partitioning" to "grid cell partitioning", which we believe is more meaningful. Hopefully our restructuring helps the reviewer, together with our replies below.

SPECIFIC COMMENTS

*The description of the setup needs careful revision. The authors do not help the reader using inconsistent terminology. I am trying to reconcile what they wrote/mean: the numerical groundwater model is based on structure grid in Cartesian coordinates (they actually state the Cartesian grid represent lat/long; how does that work?).*

We deliberately left the details on this and referred to previous work of Sutanudjaja et al. (2011) and de Graaf et al. (2015), since the lat/long approach is similar to the 5′ groundwater model of PCR-GLOBWB. However, to clarify this more we added the text (line 128):

Using such a grid means that we have to take into account for the fact that cell areas and volumes do vary in space, and therefore MODFLOW input for the recharge and the storage coefficient need to be corrected for this (see Sutanudjaja et al. (2011) and de Graaf et al. (2015) for details).

*The grid is applied over sea and land areas uniformly. Then they talk about unstructured grids. I believe they mean unstructured subdomains, because they subdivide the global model into "grids", which are actually subdomains i.e. Afro-Eurasia, Americas, Australia, Islands?! Starting on line 179, these are then termed groundwater models which are partitioned in submodels put on one core each. Load balancing is used via METIS to improve efficiency; in this process watersheds are used in the weighing.*

We note that we never used the term subdomain in the manuscript and in our terminology, we call Afro-Eurasia, Americas, Australia and Islands, *models* of the GLOBGM since they are independent models in our approach. We deliberately left out the term subdomain since we think "domain" might be nondescriptive for the readers. Instead, we use the term submodel, defined as being a part of one of the four models within the GLOBGM. We acknowledge that this could have been clearer. For that reason we introduced Section 2.1.1 (line 123) for describing the general concept, illustrating this with the new Figure 1 (e.g. line 135):

For addressing this problem, we can significantly reduce the number of grid cells by applying unstructured grids and maximize parallelism by deriving as many independent groundwater models as possible while satisfying all necessary boundary conditions. This concept is illustrated by Figure 1. Starting with the 30″ global-scale land-sea mask and boundary conditions prescribed by the GGM, we first derive independent unstructured grids and group them in a convenient way from large to small (see Section 2.1.2 for details). Then, we define the GLOBGM as a set of four independent groundwater models: three continental-scale groundwater models for the three largest unstructured grids and one remainder model called "Island model" for the remainder of the smaller unstructured grids (see Section 2.1.3 for details). The unstructured grids for these defined models are subject to parallelization: two partitioning methods (or domain decomposition methods) are considered (see Section 2.1.4 for details): one for partitioning grid cells straightforwardly (grey arrows in Figure 1) and the other for partitioning water catchments (red arrows in Figure 1). For each groundwater model, the chosen partitioning results in non-overlapping subgrids that define the computational cells for the non-overlapping groundwater submodels, where the computational work for each submodel is uniquely assigned to a processor core (MPI rank).

[Figure]

We note, also from other comments of the reviewer, that the GLOBGM definitely applies an unstructured grid for each of the four models of the GLOBGM (Afro-EurAsia, Americas, Australia, and Islands). This is now emphasised by Section 2.1.2 Procedure for deriving the independent unstructured grids, e.g. by line 191:

The resulting grids in the GLOBGM are clearly unstructured since the number of cell neighbors is not constant for all grid cells, and therefore the grid cell index cannot be computed directly: in the lateral direction, constant-head cells (Dirichlet; 0 m) near the coastal shore are not connected to any neighboring canceled sea cells, and in the vertical direction we cannot distinguish between upper and lower model layer anymore due to canceling of non-existing upper confining layer cells. Because of this, we apply the Unstructured Discretization (DISU) package with MODFLOW (Langevin et al., 2017).

*Figure 2 is not intuitive; the caption needs to be expanded.*

Sorry for being unclear here. We extensively expanded the caption (Figure 3, former Figure 2, line 277) as the reviewer suggested.

*While stating on line 156 that the study is not "fixating" (non-scientific language, revise) on speedups and scaling, this is definitely required given that the parallel implementation is the focus of this study as suggested in the title. Thus strong/weak scaling results need to be added to the transient simulation section.*

We apologize for our non-native English, and we changed "fixating" to "focusing". We agree with the reviewer on the importance of strong and weak scaling experiments. For that reason, a strong scaling was performed for the Americas model to estimate the submodel size for a given run-time target of 117 SYPD (Figure 9b, SYPD can interpreted as speedups). However, we did not perform a weak scaling analysis, for which grid refinement might be likely most appropriate instead of extending the computational domain. First, the focus of this research was on realizing a global groundwater model having a fixed 30″ resolution, and not on models having a lower or higher spatial resolution. Second, downscaling/upscaling input data and boundary conditions are generally not straightforward (e.g. drain/river packages with different drainage networks). Third, automation of the pre-processing for accommodating grid refinement would require a significant programming effort, as well as many computation hours for running and post-processing, which is beyond the scope of this paper. Fourth, the focus was here not on code profiling our MODFLOW 6 prototyping code. From previous tests, we see comparable parallel strong-scaling performance compared to our previous parallelization efforts for MODFLOW 2005 and SEAWAT. Together with the promising results with respect to SYPD (Figure 11), we did not feel the need to conduct any weak scaling experiments for this paper.

*Overall, the results with respect to SYPD appear to be promising. This raises the question, why a one/two-layer groundwater model has been implemented and applied.*

The main reason for this is the present lack of borehole and geophysical data at the global scale (see e.g. Section 4.1, bullet 2 in Condon et al. 2021). Note that we are working on collecting and interpreting more borehole/geophysical data for a future version of the GLOBGM to add more model layers to represent the geology better.

*One/two-layer groundwater flow models have been used 40 years ago. Thus, the results obtained with this type of model are at a level of sophistication that is not state-of-the-art.*

That depends on the data scarcity. For data-rich countries where many borehole/geophysical data are publicly available, like in the Netherlands (Van der Meulen et al., 2013), (regional-scale) groundwater models having one/two model layers are indeed not state-of-the-art, and we agree with the reviewer in that respect. However, when little to no subsurface data is available, which is the case for global-scale groundwater models like the GLOBGM (see our reply above), models are inherently bound to having one/two model layers. Although from a physical/modeling point of view this is undesirable, from a data-availably point of view those models can still deemed to be considered state-of-the-art.

*Especially, when figure 10 suggests that a more complex flow model is computationally possible.*

We agree that computationally there seems much more possible since we only used 12 nodes (Table 4) of the Snellius supercomputer instead of all the total ~1300 nodes, but this revelation came to the surface as our research progressed. Even with the current version of the GLOBGM, we believe performance can be further improved by adding more nodes. However, our focus was rather on usability of the current version of the GLOBGM than on computability of future versions. For future versions of the GLOBGM, we believe that the updated flow model complexity should be re-tuned with the available nodes and scenario runtimes.

*This is perhaps also the reason for the low and negative correlations in figure 14.*

Thank you for your suggestion. When looking at Figure 15 (former Figure 14), i.e. a comparison with GRACE, we only observe negative correlation for the annual correlation (Figure 15b), mainly in the Amazon and Sahara. As mentioned in the text, we think these negative correlations are more likely to be caused by large water storages in floodplains for the Amazon (line 716-717: "Furthermore, the low... floodplains during,") and higher noise-to-signal ratio in the Sahara (line 719-720: "Additionally, it is... ...like the Sahara").

*This brings up the purpose of the paper that is to show that 30′′ global transient groundwater simulations are possible. Coming from the mathematical/computational perspective, the authors solve a linear PDE on a structured grid over a domain with complex geometry.*

We unfortunately disagree with the reviewer on the comment "*the authors solve a linear PDE on a structured grid*". First, the groundwater flow equation subject to solving in our manuscript is non-linear, since river and drain (Robin/Neumann) boundary conditions are used that require (here Picard) linearization and therefore non-linear (outer) iterations. Second, the grid is unstructured and not structured (see above explanation; Section 2.1.2 in the revised manuscript). From a groundwater modeling perspective, this makes our GLOBGM one of the very first peer-reviewed presented models using the new MODFLOW 6 unstructured grid functionality.

*In total ~9.3 million cells (line 120) corresponding to DOFs are implemented.*

[Figure]

First, we apologize for making a mistake with this number the reviewer is referring to on line 120 of the first revised manuscript. This should be ~18.7 million cells, and we corrected this in the revised manuscript (line 131):

Each of the two GGM layers has 9.3 million 5′ cells (4,320 columns times 2,160 rows), and therefore the GGM has a total of ~18.7 million 5′ cells. A straightforward refinement of this grid to 30″ resolution would result in ~100 times more cells, hence 1.87 billion cells (two model layers of 43,200 columns times 21,600 rows).

Second, we disagree with the reviewer that only ~9,3 million cells / DOFs (correct: ~18.7 million) are implemented for the GLOBGM. We emphasize that the GLOBGM is not a straight-forward (nearest-neighbor) interpolated version of the 5′ GGM. Although some model input data are used at a coarser resolution using interpolation, e.g. recharge and well abstractions (see Table 1), the GLOBGM really uses detailed 30″ model input data: e.g. a 30″ model top layer derived from upscaling the 3″ MERIT DEM (see Table 1), and 30″ surface water levels derived from 30″ HydroSHEDS data (see Table 2). This makes the GLOBGM v1.0, having 278 million DOFs in total (168 million DOFs for the largest Afro-Eurasia model), sufficiently numerically challenging for HPC, stressing both the linear and non-linear solver, as well as the pre-processing for transient simulation.

*Even without disaggregation into independent groundwater models, this is not considered a large computational problem.*

First, from our estimates the serial GLOBGM would require at least 3 months of runtime (line 608: "…or 87 days runtime") to simulate 1958-2015 considering the largest Afro-Eurasia model of the GLOBGM, which is a large computational problem. Second, since the reviewer seems to refer to the transient 5′ GGM performance, even for this model runtimes are significant. Although this was not a subject of research for our paper, the GGM requires about ~50 minutes runtime in serial. Hence, for 78 years of simulation (1958-2015, 38 years + 20 years spin-up) about 78 x 50 min = 65 hours = 2.7 days is required, which is still significant.

*Thus, in terms of modeling and HPC, the study is of limited scope. The authors need to discuss why the study is relevant given the limitations outlined in the discussion above.*

We hope that our above answers have convinced the reviewer that the GLOBGM model is challenging with respect to HPC, i.e. as summarized in the conclusions of our manuscript:

- We believe that the GLOBGM is a numerically challenging, transient, groundwater model having 278 million DOFs in total (168 million DOFs for the largest Afro-Eurasia model), that we have sucessfully implemented with limited and reachable parallel hardware requirement (line 753: "successfully implemented…").
- To our knowledge, the GLOBGM is the first global-scale transient 30″ groundwater model made possible by our parallelization (line 755: "To our knowledge, …").
- The GLOBGM uses parallel pre-processing, to our knowledge new for MODFLOW-based groundwater modeling, and we showed its necessity (line 759: "using parallel pre-processing').
- The GLOBGM uses unstructured grids, making this model one of the first MODFLOW 6 models presented for peer-reviewing using this functionality (line 756: "Our implementation uses…").

[Figure]

- The GLOBGM is the first parallel application using our new parallellized version of MODFLOW 6 (line 760: "a new parallel distributed memory prototype version").
- Besides straight-forward cell partitioning, the GLOBGM can apply sub-optimal partitioning using catchments that turned out to be a promising method (769: "For catchment partitioning, …"').

Although we do believe that in terms of modeling and HPC we have significant scope and made advances, we note that these HPC advances mainly serve the primary goal of the paper (Figure 11 and 12), i.e. realizing a first transient 30″ global-scale groundwater model meeting user requirements.

REFERENCES

*Condon, L. E., Kollet, S., Bierkens, M. F. P., Fogg, G. E., Maxwell, R. M., Hill, M. C., et al. (2021). Global groundwater modeling and monitoring: Opportunities and challenges. Water Resources Research, 57, e2020WR029500. https://doi.org/10.1029/2020WR029500*

*de Graaf, I. E. M., Sutanudjaja, E. H., Van Beek, L. P. H., and Bierkens, M. F. P.: A high-resolution global-scale groundwater model, Hydrol. Earth Syst. Sci., 19, 823–837, https://doi.org/10.5194/hess-19-823-2015, 2015.*

*Sutanudjaja, E. H., Van Beek, L. P. H., De Jong, S. M., Van Geer, F. C., and Bierkens, M. F. P.: Large-scale groundwater modeling using global datasets: A test case for the Rhine-Meuse basin, Hydrol. Earth Syst. Sci., 15, 2913–2935, https://doi.org/10.5194/hess-15-2913-2011, 2011.*

*Van der Meulen, M.J., Doornenbal, J.C., Gunnink, J.L., Stafleu, J., Schokker, J., Vernes, R.W., Van Geer, F.C., Van Gessel, S.F., Van Heteren, S., Van Leeuwen, R.J.W., Bakker, M.A.J., Bogaard, P.J.F., Busschers, F.S., Griffioen, J., Gruijters, S.H.L.L., Kiden, P., Schroot, B.M., Simmelink, H.J., Van Berkel, W.O., Van der Krogt, R.A.A., Westerhoff, W.E., Van Daalen, T.M., 2013. 3D geology in a 2D country: Perspectives for geological surveying in the Netherlands. Netherlands J. Geosci. / Geol. en Mijnb. 92, 217–241. https://doi.org/10.1017/S0016774600000184*

Please let us know if you have any questions or more clarification is required. We hope that after this second round of revisions the paper has improved sufficiently to be considered for publication in GMD.

On behalf of all authors,

With kind regards,

Jarno Verkaik